# Effects of low-temperature stress at different growth stages on rice physiology, pollen viability and yield in China's cold region

Lifeng Guo[1,2], Xiaodong Du[3], Jianing Chang[4], Jingjin Gong[1,2], Zheng Chu[1,2], Jiajia Lv[1,2], Lixia Jiang[1,2]*

1 Heilongjiang Province Institute of Meteorological Science, Harbin, China, 2 Wuying National Climatological Observatory, Harbin, China, 3 Rice Research Institute of Heilongjiang Academy of Agricultural Sciences, Jiamusi, China, 4 School of Resources and Environment, Northeast Agricultural University, Harbin, China

* hljjlx@163.com

## Abstract

Low-temperature stress (LTS) is a major limiting factor for rice production in high-latitude regions. Many studies have reported the impacts of LTS on leaf photosynthesis and yield, but few of them explored the response of photosynthesis, chloroplast ultrastructure, pollen fertility, cold stress adaptation to LTS at different growth stages of rice. In this study, we conducted a two-year temperature-controlled field experiment (in 2023 and 2024) to investigate the effects of LTS at the tillering, booting, and heading stages on physiological and biochemical characteristics, plant growth, pollen fertility, and grain yield for a japonica rice cultivar (Longgeng31). The results showed that rice photosynthesis gradually decreased as the LTS temperature was decreasing and the LTS duration was increasing. The net photosynthetic rate ($P_n$) decreased the most at the booting stage, followed by the tillering, and the heading stages. Compared with controlled group (CK), the LTS treatment at 11.5°C for 3–10 days significantly reduced $P_n$ by 52.2%~62.7%, 85.3%~93.9% and 39.3%~44.9%, at the tillering stage, booting and heading stages respectively. Increasing LTS intensity and duration caused distorted chloroplast morphology and reduced plant height. The concentrations of the antioxidant and osmotic regulation systems in rice peaked after 7 days of LTS treatment, indicating that the stress response to LTS showed a trend of initially increasing and subsequently decreasing. The grain yield decreased the most under LTS at the booting stage by 59.30%−88.76% on D10, followed by the heading and tillering stages. After 10 days of exposure to LTS, the pollen viability decreased most significantly at the heading stage by 44.67%, followed by the booting and the tillering stages. These findings could provide a theoretical basis for identifying and evaluating LTS in rice under field conditions, and provide a methodological reference for the identification and monitoring of LTS in other crops, thereby holding significant practical implications.

**Data availability statement:** All relevant data are within the manuscript and its Supporting Information files.

**Funding:** This work was funded by the National Key Research and Development Program of China (No. 2022YFD2300201), the Natural Science Foundation of Heilongjiang Province (No. LH2024D020), the National Natural Science Foundation of China (No. 31671575) and the Joint Foundation on Regional Meteorological S & T Collaborative Innovation of Northeast China (No. 2024GX006). The National Key Research and Development Program of China provided financial support for personnel salaries, experimental materials, and data analysis. The Natural Science Foundation of Heilongjiang Province provided financial support for personnel salaries, experimental materials, and data analysis. The National Natural Science Foundation of China funded equipment access "Hitachi H7650" and publication fees. The Joint Foundation on Regional Meteorological S & T Collaborative Innovation of Northeast China provided financial support for personnel salaries, experimental materials, and data analysis. The funders played role in:
● Study design ● Data collection and analysis ● Decision to publish ● Preparation of the manuscript.

**Competing interests:** The authors have declared that no competing interests exist.

## Introduction

Rice (*Oryza sativa L.*) is a crucial staple crop for more than half of the global population, particularly in Asian counties [1,2]. The consumption of rice in China is essential, as it serves as a primary food source for over 65% of the population [3], despite having less than half the global average of arable land per capita [4]. The cultivation of rice spans a wide range of climates, from tropical and subtropical to temperate and even cold regions, due to its preference for specific temperature conditions [5]. Its growth has certain requirements for meteorological conditions such as water and light, and it is particularly sensitive to temperature conditions [6,7]. Low-temperature is one of the main limiting factors for rice production in high-latitude or high-altitude regions [8–10]. In recent years, under the background of global warming, the frequency of extreme weather events has changed, and the occurrence of periodic low-temperature may increase [11,12]. China, as the world's largest rice-producing country, has been seriously affected by low-temperature events [13,14], especially in the high-latitude northeast China, the continuous low-temperature in the growing season has seriously affected the growth and development of rice [15,16]. Unlike other gramineous crops, rice is more vulnerable to LTS. During the vegetative growth stage, LTS can delay development and prolong the growth period [17,18], while in the reproductive stage, it can lead to pollen sterility, an increased empty grain rate, and significant yield reduction [19]. The effect of LTS on rice production has always been the focus of rice meteorological research.

When the external environment changes, the plant's resistance to environmental stress is a physicochemical reaction process involving the coordination of multiple systems, which is not only affected by the genetic characteristics of the species, but also restricted by the environment. The LTS is a crucial abiotic stress in rice, exerting significant control over both rice production and geographical distribution [20,21]. When exposed to low-temperatures, rice cells undergo a series of morphological, physiological, and biochemical changes in their structure as well as alterations in various cellular substances to sustain normal growth. Consequently, this phenomenon primary effect of low-temperature and cold injury on rice production is the reduction in yield, primarily attributed to the reduction of anther, the increase of sterile pollen in anther, and the decrease of seed setting rate, all of which collectively affect rice productivity [22]. The effect of low-temperature injury usually destroys energy source and metabolic balance of rice [23,24], inhibits photosynthetic rate and reduces leaf area, thereby reducing plant biomass accumulation and yield [25].

The effects of low-temperature stress (LTS) on the physiological and biochemical processes of rice are complex. Under LTS, synergistic changes were observed in cell membrane stability and function, photosynthetic efficiency, antioxidant defense systems, and osmotic regulation [26,27]. Antioxidant enzymes, including peroxidase (POD), superoxide dismutase (SOD), and catalase (CAT), act as protective enzyme systems to limit the rise of free radicals and prevent them from causing damage to rice cells, and maintain the balance between antioxidant system and free radicals [28]. LTS generates excessive amounts of free radicals, peroxides, and reactive oxygen species (ROS), thereby causing damage to the antioxidant system of rice.

However, the defense mechanisms of antioxidant enzymes can effectively mitigate their harmful effects on rice [29]. The accumulation of proline (Pro) in response to LTS is an adaptive mechanism that enhances rice tolerance and can be expressed as a dynamic equilibrium between Pro accumulation and the threshold for LTS tolerance [30–32]. The accumulation of soluble proteins (SP) is also observed in response to LTS, as they possess osmotic adjustment functions and serve to protect rice cells from dehydration damage [33]. Plant hormones, including abscisic acid (ABA), jasmonic acid (JA), salicylic acid (SA), and gibberellin (GA$_3$), are among the most extensively studied due to their critical roles in regulating fundamental processes such as plant growth and development. Additionally, these hormones significantly contribute to stress responses by modulating mechanisms that enable plants to low temperature stress condition [34]. Moreover, it plays a crucial role in various processes, including the maintenance of seed dormancy and development, flowering, embryo morphogenesis, and grain filling [35].

As is well known, LTS can induce excessive accumulation of ROS, thereby seriously damaging the cell membrane composition, chloroplast ultrastructure and plastid structure in rice plants [36], restricting photosynthesis and reducing substance production [37]. The photosynthetic parameters are commonly employed as indicators of the extent of environmental stress experienced by plants. The LTS can affect the intercellular carbon dioxide concentration and stomatal conductance in rice [38]. Stomata, as a crucial conduit for of carbon assimilation in plant ecosystem, is easily restricted by various environmental and physiological factors [39]. The opening and closing of stomata in leaves play a crucial role in regulating the intercellular carbon dioxide concentration, thereby affecting photosynthesis and the biochemical processes within chloroplasts [40]. The primary limitations on photosynthesis include both stomatal and non-stomatal factors. Stomatal limitations (SL) refer to the restriction of carbon dioxide supply diffusing through stomata into the intercellular spaces of leaf tissues. Non-stomatal limitations (NSL) encompass all processes that hinder the rate of photosynthesis in leaves, excluding those related to stomata. These include the diffusion of carbon dioxide from the intercellular spaces within leaf cells to the dark reaction sites in chloroplasts, as well as various other biochemical factors that impact photosynthetic rates [41,42].

The growth of rice at each stage is dependent on a specific minimum optimum temperature, and low-temperatures hinder the accumulation of dry matter [43]. Previous studies have shown that rice is highly sensitive to low-temperatures during its reproductive stage. Exposure to these conditions can result in reduced pollen viability, decreased effective tiller count, increased empty grain rate, lowered thousand-grain weight and ultimately lead to a decline in rice yield [44,45]. The cold tolerance of various rice cultivars varies. The japonica rice cultivars originating from Korea and Japan generally demonstrate higher resistance to low-temperatures compared to those from China and the majority of Russia, for example [46,47]. The tolerance of rice to low-temperatures is not only dependent on the source or variety, but also on the growth stage, as well as the intensity and duration of the LTS [48,49]. However, the frequent occurrence of extreme weather events resulting from climate change has further exacerbated the effect of LTS on rice production [50].The need to assess and quantify the effects of low-temperature stress (LTS) on rice production stems from projected climate change scenarios and their potential implications for future agricultural productivity.

Multiple studies have been conducted to identify the growth stages that exhibit the highest susceptibility to LTS, as well as to elucidate the underlying mechanisms that account for how LTS affects rice yields. The effect of LTS on rice yield can manifest through various mechanisms [51]. During the seedling stage, LTS can delay rice development and even cause seedling mortality [52], while during the vegetative growth stage, it reduces yield by decreasing biomass accumulation [49]. During the jointing stage, LTS decreases rice yield by reducing the number of spikelets per panicle [53]. During the booting stage, the exposure to LTS can lead to the occurrence of male sterility, a phenomenon that subsequently leads to a significant decrease in the number of spikelets within each panicle. The reproductive characteristics of rice are ultimately influenced by the direct effect of LTS, thereby affecting the final yield and quality [54,55]. During the heading and flowering stage, LTS can hinder the natural dehiscence of anthers, subsequently compromising the process of pollination. As a result, this leads to a significant decrease in pollen fertilization rates and a substantial increase in the proportion of empty hulls [53,56].

The impact of LTS on rice cultivated under natural field conditions was significantly distinct from that observed in rice grown under laboratory-controlled conditions [57]. Previous studies on rice LTS have predominantly focused on constant low-temperature conditions or limited temperature gradient treatments during specific growth stages. And many studies have reported the impacts of LTS on leaf photosynthesis and yield, but few of them explored the response of photosynthesis, chloroplast ultrastructure, pollen fertility, cold stress adaptation to LTS at different growth stages of rice [22,58]. However, hourly dynamic low-temperature simulation based on natural diurnal rhythms has not been adequately addressed in the existing literature. Therefore, this study designed and implemented a rice LTS experiment replicating the low-temperature conditions experienced under natural field conditions. Based on a two-year LTS experiment, with rice growth and development under natural conditions serving as the control, this study comparatively analyzed the effects of hourly dynamic LTS on growth, physiological and biochemical characteristics, and yield at different growth stages of rice. Among all cereals, rice is highly sensitive to LTS, such as the seed germination stage, which adversely impacts its germination ability, seed vigor, crop stand establishment, and, ultimately, grain yield [59,60]. Meanwhile, the impact of LTS plays a crucial role in the breeding process of high-quality japonica rice [61]. Moreover, LTS constitutes one of the critical factors influencing the accuracy of rice climate yield predictions generated by crop models [62].The findings of this study could provide a theoretical basis for identifying and evaluating LTS in rice under field conditions, and provide a methodological reference for the identification and monitoring of LTS in other crops, thereby holding significant practical implications.

## Materials and methods

### Experimental design

Controlled-environment LTS phytotron experiments were conducted in 2023 and 2024 at Rice Research Institute of Heilongjiang Academy of Agricultural Sciences in Jiamusi (46.87°N and 130.53°E), Heilongjiang Province, China. A Japonica inbred rice cultivar, Longgeng31, with moderate cold tolerance was selected for the experiment. Seedlings at the three-leaf stage were transplanted into plastic pots (28 cm in diameter and 45 cm in height), each filled with 15 kg of dry paddy soil sourced from the experimental farm of the Rice Research Institute at the Heilongjiang Academy of Agricultural Sciences. The soil's basic physical and chemical properties were as follows: pH 6.22, organic matter content 36.56 g/kg, alkali hydrolysis nitrogen content 105.35 mg/kg, available phosphorus content 82.56 mg/kg, and available potassium content 92.6 mg/kg. After undergoing disinfection, seed soaking, and germination, the seeds of Longgeng31 were sown on seedling plates containing paddy soil and cultivated in a plastic shed. Once the rice seedlings reached the three-leaf and one-heart leaf stage, seedlings with similar developmental stages were carefully selected and transplanted into the plastic pots. The transplanting standards were kept consistent with those of the RILs. Field management practices were done according to the most followed agricultural practices of local farmers. The planting density was 3 hills per pot (3 seedlings per hill) for a total of 90 pots. All the pots were grown under ambient weather conditions before LTS treatment. 2 g ammonia sulphate, 1 g potassium sulphate, and 1 g diammonium phosphate were applied in each pot as the basal fertilizer, and an additional 1 g ammonia sulphate was applied at the panicle differentiation stage of rice. The pots were kept flooded until one week before harvesting. Watering at the late active tillering stage was stopped to ensure efficient tillering. Weeds were removed manually, whereas pest and diseases were controlled using pesticides.

Thirty-six pots of each low-temperature treatment were transferred into the two phytotrons respectively, once the rice plants reached the targeted tillering, booting, and heading stages. LTS was designed at two temperature levels of T2 and T3 (with daily average temperature at 13.5°C and 11.5°C, respectively) at tillering stage, T1 and T2 (with daily average temperature at 17.5°C and 13.5°C, respectively) at booting and heading stages, and three temperature durations of D3, D7, D10 (3days, 7days, and 10 days, respectively), with 12 h-light/12 h-dark photoperiod, 70% RH. This experiment adopted a 24-hour variable temperature setting. The hourly temperature for each LTS treatment was shown in Fig 1.

In this study, rice subjected to low-temperature treatment was compared with rice grown under natural conditions, which served as the control group (CK). This comparison aimed to analyze the effects of varying intensities of LTS on the

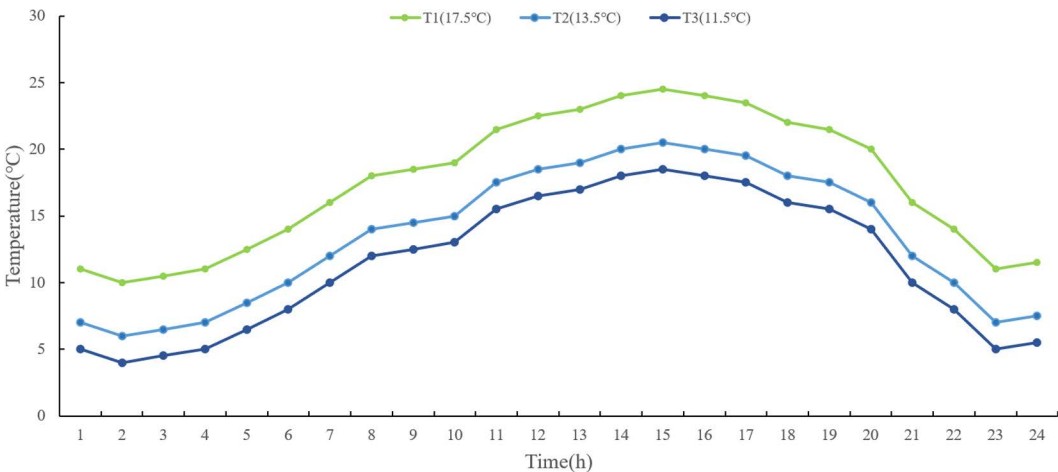

**Fig 1. The hourly temperatures in the phytotron.**

growth and development of different rice growth stages. The average daily mean temperature for rice cultivated under natural conditions in 2023 and 2024 was utilized as the reference average temperature for the CK in this study, and the temperature conditions for each CK are presented in Table 1.

Upon the completion of each low-temperature treatment for the designated duration, twelve pots of the rice were transferred outdoors to resume growth under natural conditions until they achieved full maturity.

### LTS treatment and sample preparation

At the tillering stage, when the tillering ratio of rice plants reached 50%, 12 pots exhibiting uniform growth were randomly selected and transferred to individual phytotrons for LTS treatment. The LTS treatment applied during the booting and heading stages was identical to that administered during the tillering stage. After subjecting the pots to LTS treatment for 3, 7, and 10 days (12 h-light/12 h-dark photoperiod, 80% RH) at each growth stage, they were subsequently transferred to an outdoor natural environment for cultivation until reaching maturity. The fully expanded flag leaf samples at the top were collected from each treatment, with a total weight of 0.5 g per sample. The sampling process was repeated three times from different pots to ensure the reliability of sample collection. The fresh flag leaf samples were promptly flash-frozen in liquid nitrogen and subsequently stored at a temperature of −80°C.

### Yield and its components

After the plants reached maturity, nine hills rice plants (3 pots) were randomly selected from each treatment. The plants were harvested from the base on the soil surface. The sample plants were first separated into green leaves, dead leaves, stems and panicles and then dried at 80°C for 72 h or until the weight stopped decreasing. The harvested plants from each replication were used to determine grain yield and yield components. Grain Yield (g per plot) and yield components as 1000-grain weight (g), Seed set percentage (%), grain number per panicle, and effective panicles (number per pot) were calculated after harvesting at physiological maturity as previously described [63].

### Photosynthetic parameters

The photosynthetic characteristics of the flag leaf were measured within 2–3 days after each LTS treatment, with a total of 9 repetitions per treatment. The changes of net photosynthetic rate ($P_n$), intercellular $CO_2$ concentration ($C_i$), stomatal

**Table 1. The average daily mean temperature, average daily minimum temperature, and average daily maximum temperature of each CK group of cultivated rice grown under natural conditions in 2023 and 2024.**

| Growth stage | Duration of LTS treatment | Average daily mean temperature(°C) (CK) | Average daily minimum temperature(°C) | Average daily maximum temperature(°C) |
|---|---|---|---|---|
| Tillering stage | D3 | 18.2 | 15.1 | 21.2 |
| | D7 | 20.4 | 15.8 | 24.8 |
| | D10 | 21.2 | 16.9 | 25.7 |
| Booting stage | D3 | 19.9 | 16.0 | 24.2 |
| | D7 | 23.3 | 18.4 | 28.2 |
| | D10 | 23.6 | 19.0 | 28.1 |
| Heading stage | D3 | 24.3 | 19.5 | 29.0 |
| | D7 | 24.7 | 20.3 | 29.4 |
| | D10 | 23.9 | 20.4 | 28.1 |

conductance ($G_s$) and transpiration rate ($T_r$) on the fully expanded flag leaves of rice plants under different LTS treatments were measured using a portable type Li-6400 (Li-6400, Li-Cor Inc., USA) plant photosynthetic measurement apparatus between 9:30–11:30. The analyses were performed under an air flow rate of 200μmol·s−1, 65% humidity, 350μmol (photon)·m$^{-2}$·s$^{-1}$ light intensity, and about 350 μmol·mol−1 ambient $CO_2$ concentration [64]. The analyses were conducted under an air flow rate of 200 μmol·s$^{-1}$, a humidity level of 65%, a light intensity of 350 μmol (photon)·m$^{-2}$·s$^{-1}$, and an ambient $CO_2$ concentration of approximately 400 μmol·mol$^{-1}$.

## Determination of the content of ROS and lipid peroxides

The MDA content was determined by the reaction with thiobarbituric acid (TBA) [65]. The absorbance of MDA was measured at 440, 532, and 600 nm using UV–Vis Spectrophotometer (Mettler-Toledo UV5Bio, Switzerland).

The superoxide ($O_2^-$) content was determined following the method of Batool et al [66]. The phosphate buffer (pH 7.8, 0.5 mL) was mixed with p-aminobenzene sulfonic acid (17 mM, 1 mL), hydroxylammonium chloride (1 mM, 1 mL), and α-naphthylamine (7 mM, 1.0 mL). The reaction mixture was incubated at a temperature of 25°C for a duration of 60 minutes and subsequently measured for absorbance at a wavelength of 530 nm.

The $H_2O_2$ content was determined as described by Song et al [67]. The samples were homogenized with trichloroacetic acid (0.1%) and subsequently centrifuged at 10,000×g at 4°C for 20 minutes. Afterwards, 1 mL of the resulting supernatant was combined with 2 mL of KI (1 M) and 1 mL of $K_2PO_4$ buffer, followed by measuring the absorbance at a wavelength of 390 nm after subjecting to darkness treatment for one hour.

EL was calculated by referring to the experimental analysis method of Li et al [68]. Each fresh 0.5 g leaf of rice was placed in individual stoppered triangular flasks containing 10 mL of deionized water. The samples were incubated at 25°C for 24 h. The electrical conductivity of the solution (S1) was measured after incubation. The samples were then placed in a boiling water bath for 10 min and the second measurement (S2) was determined after cooling the solutions to room temperature. EL＝S1/S2×100%.

## Determination of antioxidant enzyme activity

The frozen flag leaf samples were homogenized with a pre-cooled pestle and mortar in potassium phosphate buffer (50 mM, pH 7.8) supplemented with 1% polyvinylpyrrolidone. The prepared crude enzyme extract was incubated with the supernatant obtained after homogenization and subsequently centrifuged at 15,000×g at 4°C for 20 min. The activities of POD, SOD, and CAT were measured following the method of Xu et al [69].

## Determination of the contents of osmotic adjustment substances

The estimation of free proline was conducted according to the method proposed by Bates et al. with minor modifications [70]. The samples were immersed in 5 mL of sulphosalicylic acid (3%, 100°C for 10 minutes). Subsequently, a volume of 2 mL from the resulting extract was combined with ninhydrin reagent containing glacial acetic acid and incubated at 100°C for a duration of 30 minutes. After being cooled in ice water, 4 mL of toluene was added to the mixture and subsequently measured at a wavelength of 520 nm for the determination of proline content. The quantification of SP content was performed using the BCA method according to the established protocol by Campion et al [71]. The quantification of soluble sugar content was determined following the experimental analysis protocol established by Loutfy et al [72].

## Preparation of chloroplast electron microscope samples

The first wholly unfolded leaf at the top of the rice of different treatments was cut into 2 mm$^2$ miniature pieces with a sharp double-sided knife (at the same position on the leaf) and placed in a refrigerator at 4°C with 2.5% glutaraldehyde for pre-fixation. The electron microscope samples, fixed with glutaraldehyde, were subjected to three rounds of cleaning with PBS (0.1 mol/L, pH 6.8) at 15-minute intervals. After fixation with osmium tetroxide for 1.4 hours, the samples were subjected to three rounds of PBS cleaning for 10 minutes each. Following the cleaning process, the samples underwent dehydration using a gradient of ethanol concentrations (50%, 70%, 90%, and 100%), with each concentration being applied twice for a duration of 10 minutes per application. The dehydrated samples were immersed in a 1:1 mixture of 100% ethanol and 100% acetone for a duration of 10 minutes, followed by immersion in pure acetone for 5 minutes. Subsequently, the samples were soaked in a solution containing 100% acetone and Epon812 resin (in a ratio of 1.0:1.5) for an hour. After complete embedding in pure resin at a temperature of 30°C for 5 hours, the polymerization process was initiated with a gradual increase in temperature to reach between 36–48 hours at 60°C. Finally, the samples were embedded using pure embedding agent until the following day and sliced using an LKB5 micro-slicing machine before being observed and photographed under Hitachi H7650 transmission electron microscopy [73]. Transmission electron microscopy (TEM) analysis of leaves was carried out following a previous method [74].

## Determination of the pollen fertility

On the day of rice flowering, three representative panicle spikelets from three different panicles were selected from each treatment and three anthers were collected from each spikelet for analysis. The flowers were stained with I$_2$-KI solution and placed under a stereomicroscope to observe the staining of three visual fields and determine the fertility of pollen grains.

## Statistical analyses

In the experiment, the plants in the pots were treated with artificial LTS at two levels (T2 and T3, T1 and T2), and three durations (D3, D7 and D10) during the tillering stage, as well as the booting and heading stages, in two phytotrons respectively. A Completely Randomized Design (CRD) with a factorial treatment structure was employed. Homogeneous rice plants at the target growth stage were randomly assigned to all combinations of LTS levels (T1, T2 and T3) and exposure durations (D3,D7 and D10). Each unique LTS level and duration combination constituted a distinct treatment. For each growth stage studied, this randomization process was performed independently on a fresh cohort of plants. Treatments were applied concurrently within phytotrons, with 12 biological replicates (pots) per treatment combination. The average of two years' test results for each treatment was selected for analysis to ensure the repeatability and reliability of the LTS test outcomes. All the data obtained were statistically analyzed using the SPSS 27.0 (SPSS Inc., Chicago, IL, USA) for variance (ANOVA) analysis. The least significant difference (LSD) test was used to determine the significant differences among treatments (P < 0.05).

 

## Results

### Effect of LTS on photosynthetic enzymes

The LTS treatment at the tillering, booting, and heading stages reduced $P_n$ of rice leaves (Fig 2). $P_n$ showed a decrease as the LTS temperature decreased and the LTS duration increased. At T3, $P_n$ was most reduced by LTS treatment at booting stage, followed by the tillering stage, and the heading stage. At tillering stage, with the increase in LTS duration, the LTS treatment at T2 and T3 significantly reduced $P_n$ by 52.2%~62.7%, and 65.9%~78.3% compared with CK, respectively (Fig 2a). At booting stage, there was no significant decrease in $P_n$ when the LTS duration was increased at T1 from D3 to D7, but $P_n$ showed a significant decrease by 29.6% of D10. And, there was significant decrease in $P_n$ when the LTS duration was increased at T3 from D7 to D10, by 85.3%, and 93.9% compared with CK, respectively (Fig 2b). At heading stage, with the LTS duration increased, $P_n$ showed a non-significant decrease at T1, but a significant decrease at T3 by 39.3%~44.9% compared with CK (Fig 2c).

$C_i$ increased under the LTS treatment, and its became more pronounced as the LTS temperature decreased (Fig 3). At the tillering stage, $C_i$ initially increased and subsequently decreased as the duration of LTS treatment extended, and the increase of $C_i$ reached the highest at T3 of D3, which was preeminently resulted in the higher accumulation of 25.26% relative to control (Fig 3a). At booting stage, $C_i$ elevated gradually with the decrease of treatment temperature while the duration of low-temperature treatment was the same. $C_i$ of T3 treatment was 4.97%~39.88% higher than that of T1 treatment (Fig 3b). There was no effective change in $C_i$ of 3d leaves treated by T1 at heading stage, while there was noted difference in other treatments compared to those in the control treatments at the same time. Compared with CK, $C_i$ was memorably elevated by 13.82% and 19.91% at each temperature for the duration of D10 (Fig 3c).

Under LTS, with the decrease of temperature, $G_s$ showed a decreasing trend under all LTS treatments, and the decrease was most obvious at T3. At T3, with the increase of LTS days, $G_s$ decreased by 65.88%~86.10%, 22.59%~81.00% and 20.94%~67.71%, respectively, compared with CK in the same period (Fig 4). At the tillering stage, $G_s$ initially decreased and then increased with the duration of LTS treatment days, and most significant decrease of $G_s$ occurred on the 7th day compared to the CK (Fig 4a). At the booting stage from 7 to 10 days, with the decrease of low-temperature, $G_s$ decreased gradually and the difference between treatments was significant (Fig 4b). The same performance was also observed when LTS treatment was carried out for 3 days at heading stage (Fig 4c).

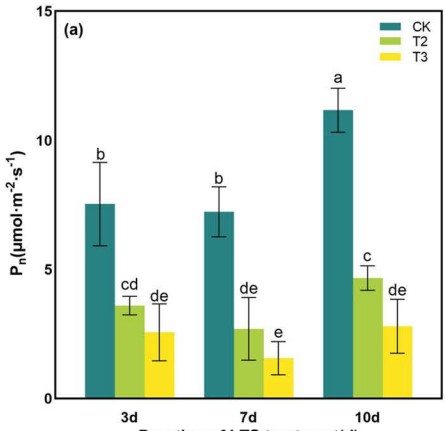 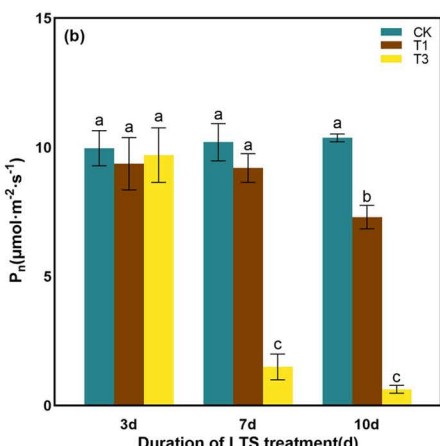 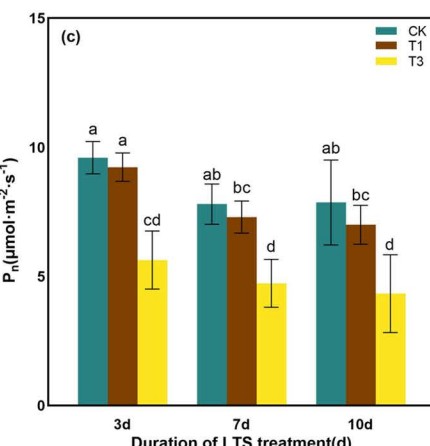

**Fig 2. Effects of LTS on $P_n$.** (a)-(c) the variation trend of the tillering, booting and heading stages. CK, under natural conditions (control); T1, under LTS of 17.5°C; T2, under LTS of 13.5°C; T3, under LTS of 11.5°C. 3d, 7d and 10d were duration of LTS. Error bars represented Mean±SE (n=3). Different letters in lowercase indicated significant difference of the data in all treatments at $P<0.05$.

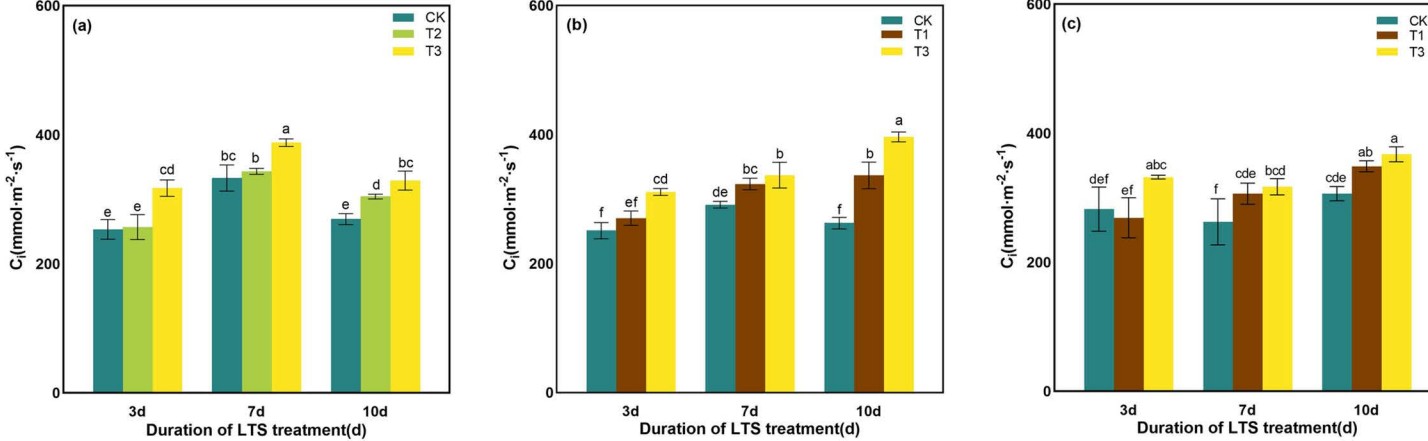

**Fig 3. Effects of LTS on $C_i$.** (a)-(c) the variation trend of the tillering, booting and heading stages. CK, under natural conditions (control); T1, under LTS of 17.5°C; T2, under LTS of 13.5°C; T3, under LTS of 11.5°C. 3d, 7d and 10d were duration of LTS. Error bars represented Mean±SE (n=3). Different letters in lowercase indicated significant difference of the data in all treatments at $P<0.05$.

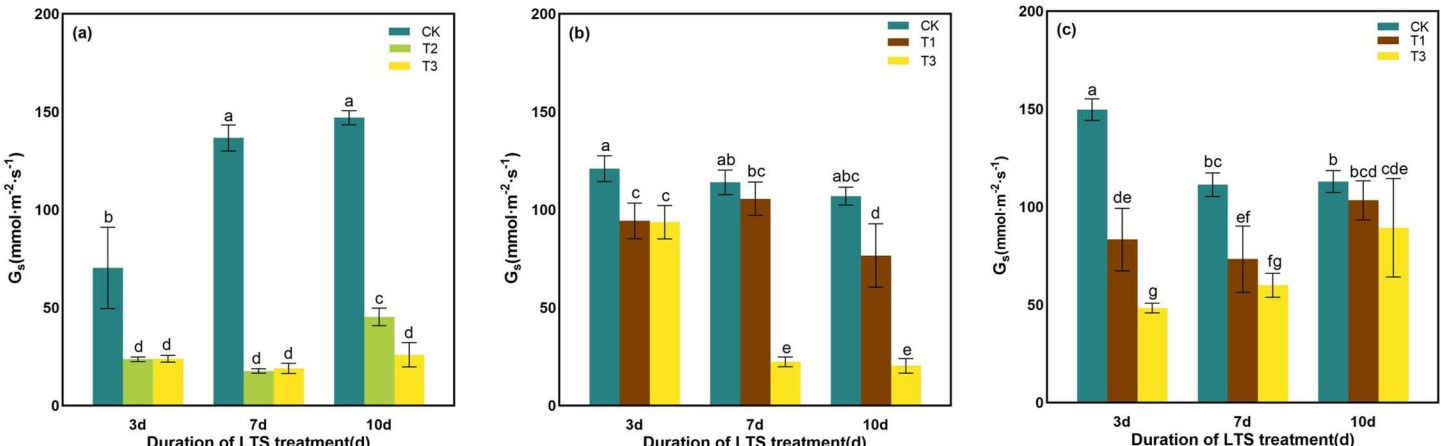

**Fig 4. Effects of LTS on $G_s$.** (a)-(c) the variation trend of the tillering, booting and heading stages. CK, under natural conditions (control); T1, under LTS of 17.5°C; T2, under LTS of 13.5°C; T3, under LTS of 11.5°C. 3d, 7d and 10d were duration of LTS. Error bars represented Mean±SE (n=3). Different letters in lowercase indicated significant difference of the data in all treatments at $P<0.05$.

At tillering stage, $T_r$ decreased by low-temperature, and it decreased by 42.04%~68.75% and 37.70%~78.13%, respectively, compared with CK treatment (Fig 5a). At booting stage, the $T_r$ of leaves gradually decreased with the decrease of treatment temperature under the same treatment days. In comparison to control plants, $T_r$ was spanking reduced by 47.32% and 84.94% after 10d of LTS treatment (Fig 5b). At heading stage, the $T_r$ of rice leaves increased first and then decreased with the increase of LTS treatment duration. The LTS treatment at T1 strikingly reduced $T_r$ by 33.33% compared with the LTS treatment at T3, for the LTS treatment duration of D3 (Fig 5c).

### Effect of LTS on oxidative stress indicators

MDA content of leaves changed significantly after low-temperature treatments. The content of MDA increased with the decrease of treatment temperature and the extension of treatment time, and T3 treatments prompted to maximum MDA

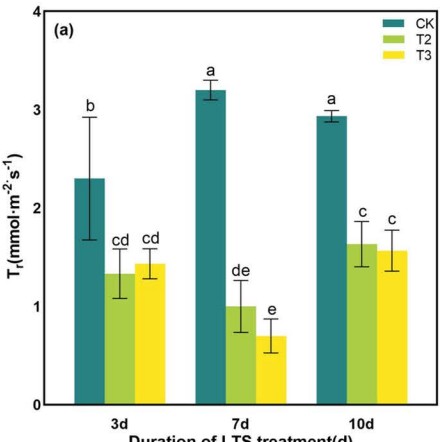 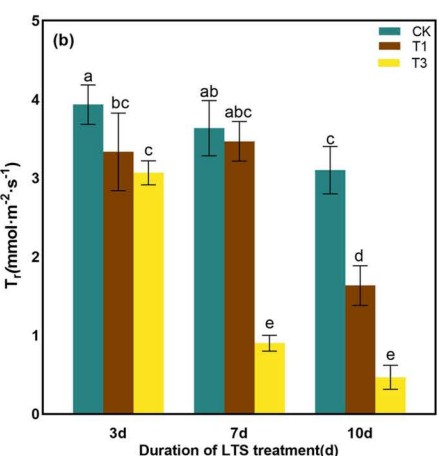 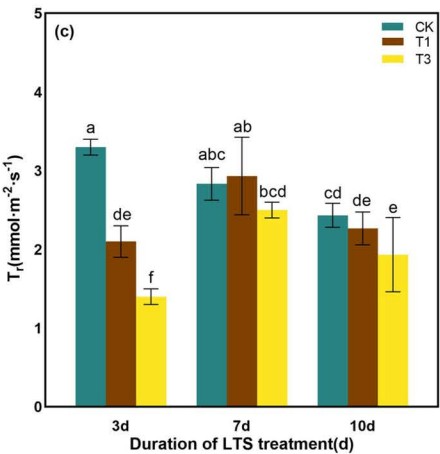

**Fig 5. Effects of LTS on T$_r$.** (a)-(c) the variation trend of the tillering, booting and heading stages. CK, under natural conditions (control); T1, under LTS of 17.5°C; T2, under LTS of 13.5°C; T3, under LTS of 11.5°C. 3d, 7d and 10d were duration of LTS. Error bars represented Mean±SE (n=3). Different letters in lowercase indicated significant difference of the data in all treatments at $P<0.05$.

content at 10 days. Compared with CK, MDA content at tillering, booting and heading stages for D10 was notedly elevated by 119.87%, 88.37% and 84.09% under T3 treatments, respectively (Fig 6).

EL of rice leaves was much exacerbated when plants were exposed to low-temperature. Compared with normal temperature treatment, EL particularly increased by 36.51%, 61.81% and 95.04%, respectively, under T2 treatment on 3~10 days at tillering stage (Fig. 7a). After 10 days of treatment, the maximum value was 2.14 times, 2.41 times and 2.36 times of the control treatment under T3 treatments at tillering, booting and heading stages, respectively (Fig 7).

LTS promoted the accumulation of $O_2^-$ in rice, and the production rate of $O_2^-$ in the plants treated with low-temperature was always higher than that in the CK. The production rate of $O_2^-$ increased gradually with the extension of stress time. Compared with the CK, the production rate of $O_2^-$ was significantly increased by 91.21%, 66.88% and 42.48% after 10

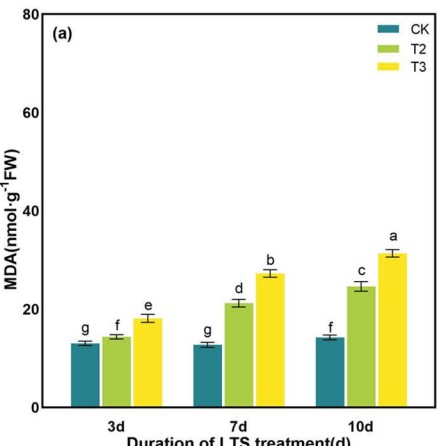 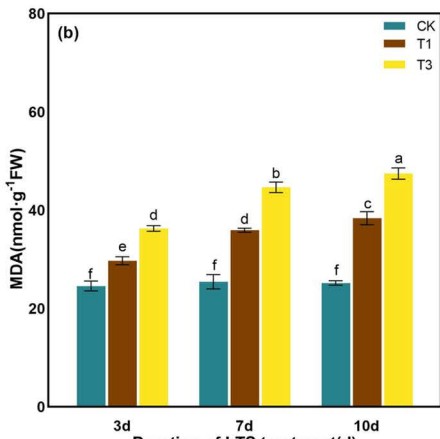 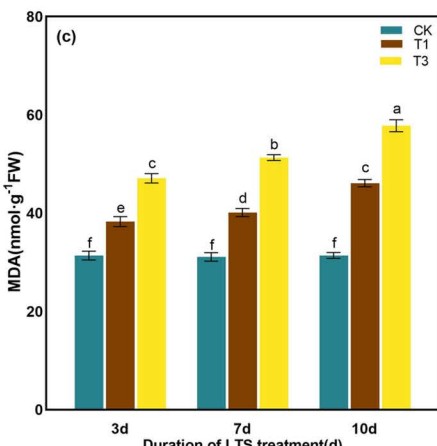

**Fig 6. Effects of LTS on MDA.** (a)-(c) the variation trend of the tillering, booting and heading stages. CK, under natural conditions (control); T1, under LTS of 17.5°C; T2, under LTS of 13.5°C; T3, under LTS of 11.5°C. 3d, 7d and 10d were duration of LTS. Error bars represented Mean±SE (n=3). Different letters in lowercase indicated significant difference of the data in all treatments at $P<0.05$.

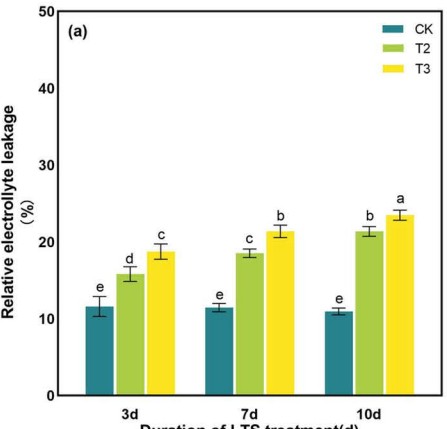
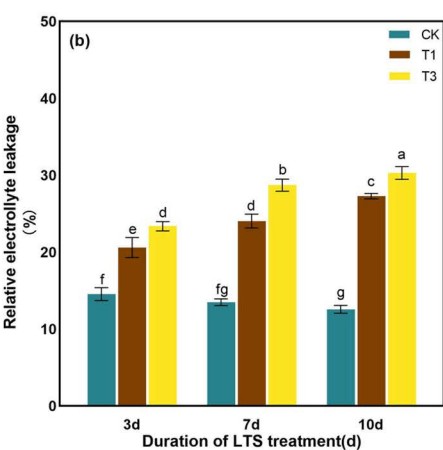
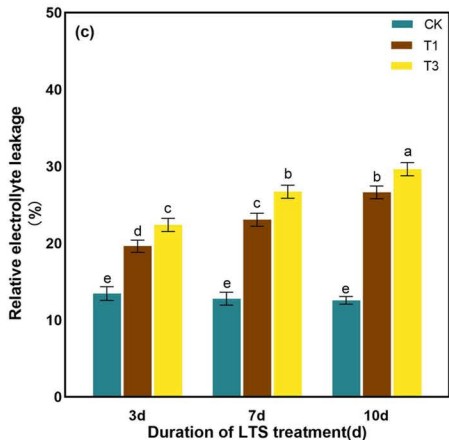

**Fig 7. Effects of LTS on EL.** (a)-(c) the variation trend of the tillering, booting and heading stages. CK, under natural conditions (control); T1, under LTS of 17.5°C; T2, under LTS of 13.5°C; T3, under LTS of 11.5°C. 3d, 7d and 10d were duration of LTS. Error bars represented Mean±SE (n=3). Different letters in lowercase indicated significant difference of the data in all treatments at *P*<0.05.

days of T3 low-temperature treatment at tillering, booting and heading stages, respectively. It was significantly higher than 10.19%, 7.30% and 6.32% on 7d (Fig 8).

The change of $H_2O_2$ content in different periods under LTS was measured. The results showed that the $H_2O_2$ content of rice under LTS treatments aggrandized significantly from 3 to 10 days, and was significantly higher than that of the CK. With the advance of stress time, the change of $H_2O_2$ content showed an increasing trend, and reached the peak value at 10 days after T3 treatment, which showed an increase of 244.55%, 122.09%, and 137.03% at tillering, booting and heading stages, respectively, compared to the values under no stress conditions. Under the LTS, the $H_2O_2$ content increased the most at D10 (Fig 9).

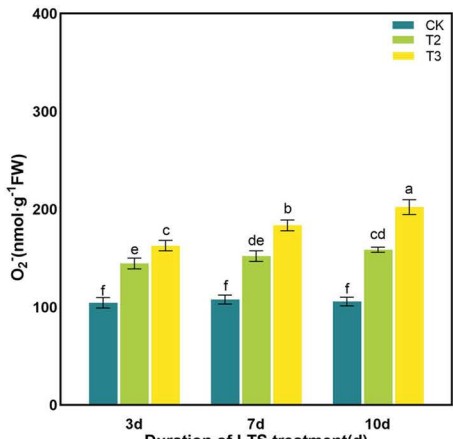
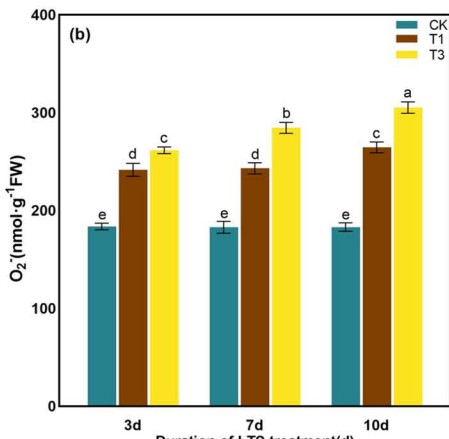
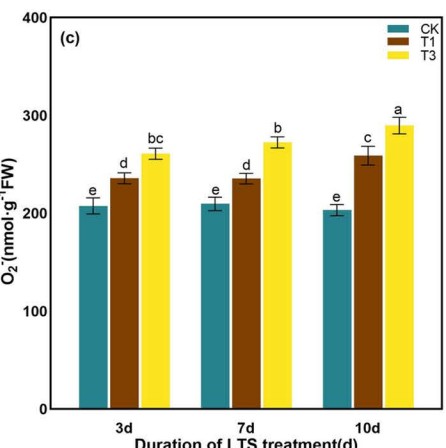

**Fig 8. Effects of LTS on the production rate of $O_2^-$.** (a)-(c) the variation trend of the tillering, booting and heading stages. CK, under natural conditions (control); T1, under LTS of 17.5°C; T2, under LTS of 13.5°C; T3, under LTS of 11.5°C. 3d, 7d and 10d were duration of LTS. Error bars represented Mean±SE (n=3). Different letters in lowercase indicated significant difference of the data in all treatments at *P*<0.05.

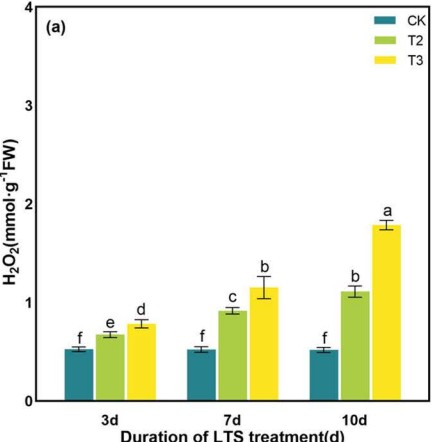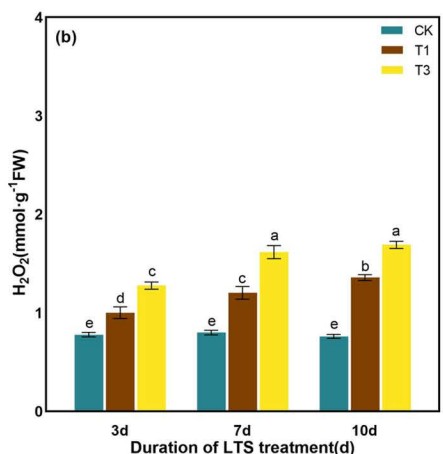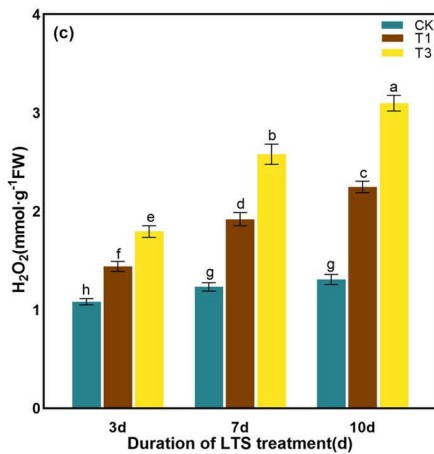

**Fig 9. Effects of LTS on H₂O₂.** (a)-(c) the variation trend of the tillering, booting and heading stages. CK, under natural conditions (control); T1, under LTS of 17.5°C; T2, under LTS of 13.5°C; T3, under LTS of 11.5°C. 3d, 7d and 10d were duration of LTS. Error bars represented Mean±SE (n=3). Different letters in lowercase indicated significant difference of the data in all treatments at *P<0.05*.

### Effect of LTS on antioxidant enzyme activity

As shown in Fig 10, rice leaves maintained a higher accumulation of SOD activity at the tillering stage at T2 and T3, and augmented conspicuously compared with that at room temperature. Under the same duration of the LTS at booting stage and heading stages, SOD activity of rice leaves could be supplemented with the decrease of temperature, compared with normal level. In different time stages, SOD activity in rice leaves increased first and then decreased with the extension of time, and reached the maximum value on the 7th day under T3 treatment, which a particularly improvement by 17.11%, 15.12% and 19.31% relative to those in the non-stressed plants, respectively.

The POD activity of rice leaves showed the trend of "unimodal curve", and reached the peak value on the 7th day after the LTS treatments (Fig 11). At tillering stage, with the comparison of CK, POD activity was significantly elevated by 5.81%~9.91% at D3~D10 under T2, while it was significantly elevated by 12.77%~17.50% at D3 and D7 under T3,

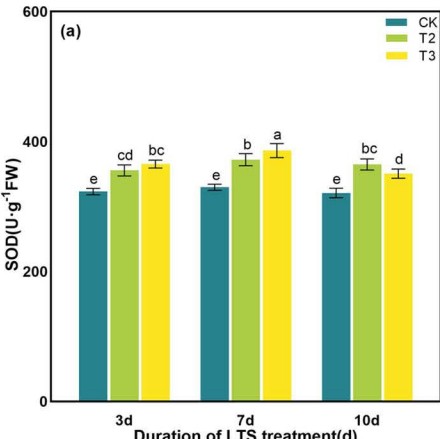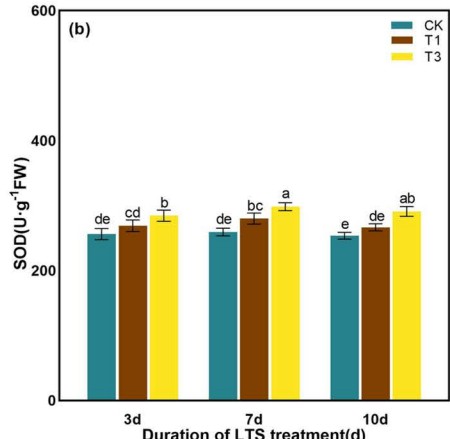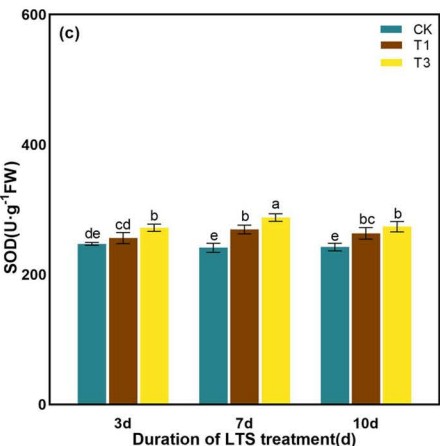

**Fig 10. Effects of LTS on SOD.** (a)-(c) the variation trend of the tillering, booting and heading stages. CK, under natural conditions (control); T1, under LTS of 17.5°C; T2, under LTS of 13.5°C; T3, under LTS of 11.5°C. 3d, 7d and 10d were duration of LTS. Error bars represented Mean±SE (n=3). Different letters in lowercase indicated significant difference of the data in all treatments at *P<0.05*.

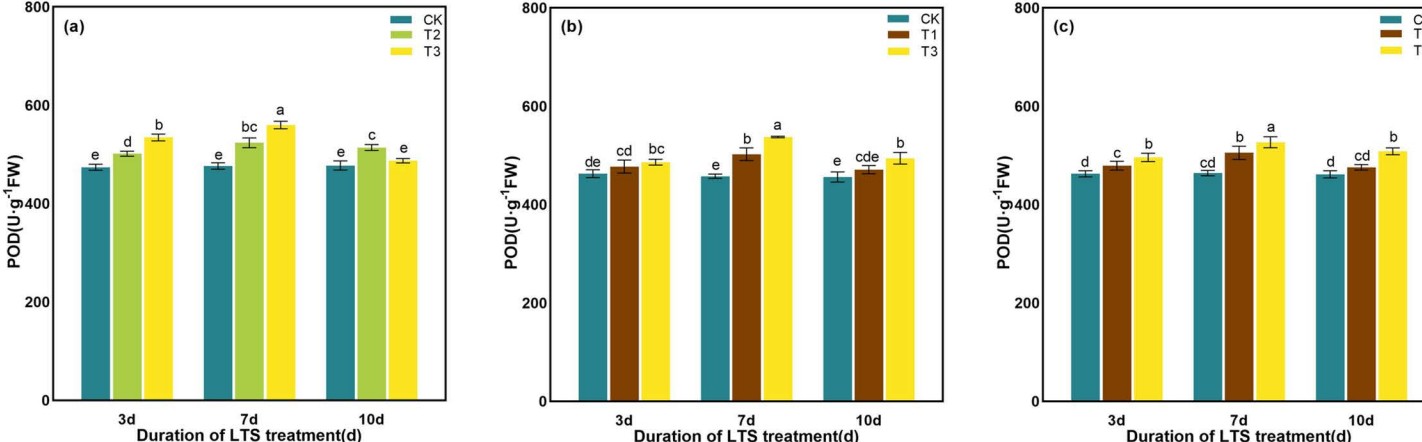

**Fig 11. Effects of LTS on POD.** (a)-(c) the variation trend of the tillering, booting and heading stages. CK, under natural conditions (control); T1, under LTS of 17.5°C; T2, under LTS of 13.5°C; T3, under LTS of 11.5°C. 3d, 7d and 10d were duration of LTS. Error bars represented Mean±SE (n=3). Different letters in lowercase indicated significant difference of the data in all treatments at *P<0.05*.

but non- significantly changed at D10 (Fig 11a). At booting and heading stages, the POD activity showed similar changes under T3. The POD activity was significantly elevated by 5.03%~17.42% and 7.20%~13.49% at D3~D10, as compared with the CK (Figs 11b and 11c).

At tillering stage, the CAT enzyme activity of rice leaves showed a notable decline trend after 10 days of T2 and T3 treatments, which decreased by 11.36% and 27.32%, respectively, as compared to those in the control treatments (Fig 12a). Under the same low-temperature treatment, the increase of CAT enzyme activity after 7 days of low-temperature treatment was higher than that after 3 days and 10 days, which significantly increased by 13.11% and 21.74%, 16.12% and 28.81%, 14.97% and 23.77, respectively, as compared with the CK in the same period (Fig 12).

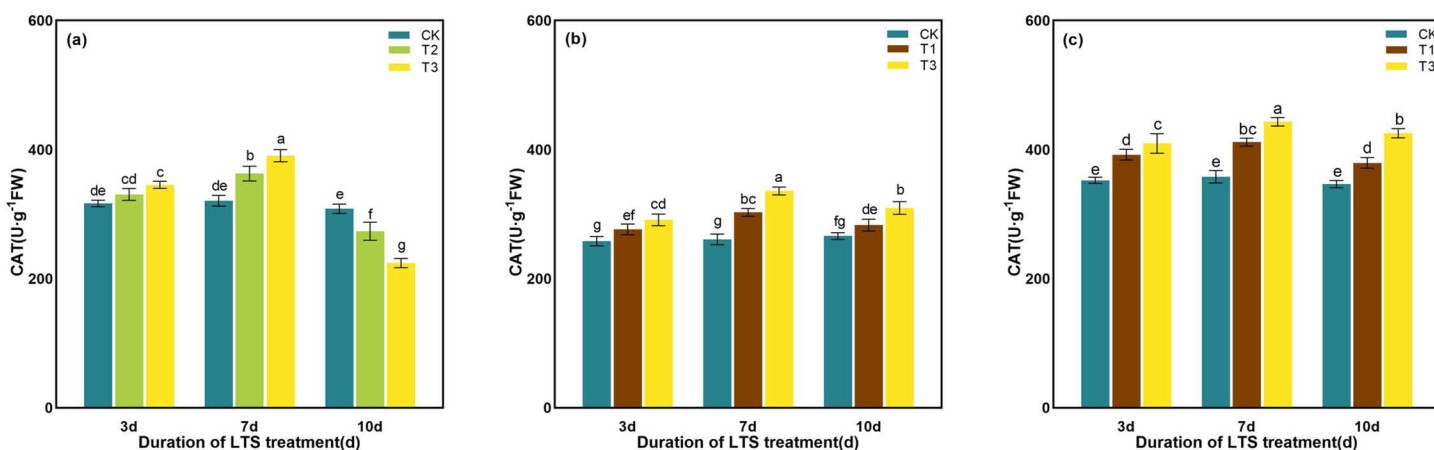

**Fig 12. Effects of LTS on CAT.** (a)-(c) the variation trend of the tillering, booting and heading stages. CK, under natural conditions (control); T1, under LTS of 17.5°C; T2, under LTS of 13.5°C; T3, under LTS of 11.5°C. 3d, 7d and 10d were duration of LTS. Error bars represented Mean±SE (n=3). Different letters in lowercase indicated significant difference of the data in all treatments at *P<0.05*.

## Effect of LTS on osmolyte contents

LTS resulted in the higher accumulation of proline in rice at different stages. However, the proline content in rice leaves showed no significant alteration under LTS of the T2 on D3 at the tillering stage, the T1 on D10 at the booting stage, and the T1 on D3 at the heading stage. Under the same low-temperature treatment, the proline content increased first and then decreased with the extension of days, reaching a peak on D7. In the same days of treatment, the proline content increased gradually with the decrease of temperature, except for 10 days of T3 treatment at tillering stage (Fig 13). For example, during the booting stage, the proline content progressively increased as the temperature declined. Notably, the increase under T3 was significantly higher than that under T1 compared with CK, and both reached the peak on D7. The content of proline preeminently accumulated by 13.10% and 48.66% in comparison with CK on D7 (Fig 13b). Proline showed a similar changing trend under LTS during the tillering and the heading stages (Figs 13a and 13c).

With the extension of low-temperature treatment time, the soluble sugar content of rice showed a unimodal curve in different growth stages. The maximum value was reached after 7 days of LTS treatment. The variation in soluble sugar content differed across durations of LTS. The content of soluble sugar was spanking higher after 7 days of low-temperature treatment at the booting stage than other treatments, which was particularly increased by 59.25% and 100.52% compared with the CK, respectively (Fig 14).

As the temperature decreased under LTS, the content of soluble protein progressively increases. Furthermore, with the prolonged duration of LTS, the content of soluble protein exhibited an initial increase followed by a subsequent decrease, peaking on D7 (Fig 15). At the tillering stage, under LTS at T2 and T3 for D3, D7 and D10, the content of soluble protein increased by 1.56%, 2.66%, 2.65% and 2.11%, 5.54%, 1.57%, compared with CK, respectively (Fig 15a). Soluble protein content increased during both booting and heading stages under LTS (T1 and T3) compared with the CK, with increments of 3.09%, 7.74%, 4.98% (T1) and 4.70%, 12.06%, 4.27% (T3) at D3, D7, and D10 at the booting stage, and 2.05%, 7.26%, 3.69% (T1) and 6.03%, 9.42%, 6.68% (T3) at the same durations at heading stage (Figs 15b and 15c). The contents of soluble protein in rice leaves reached the maximum value at the duration of D7 under T3, increasing by 5.54%, 12.06% and 9.42% compared with CK at different growth stages, respectively ($P < 0.001$).

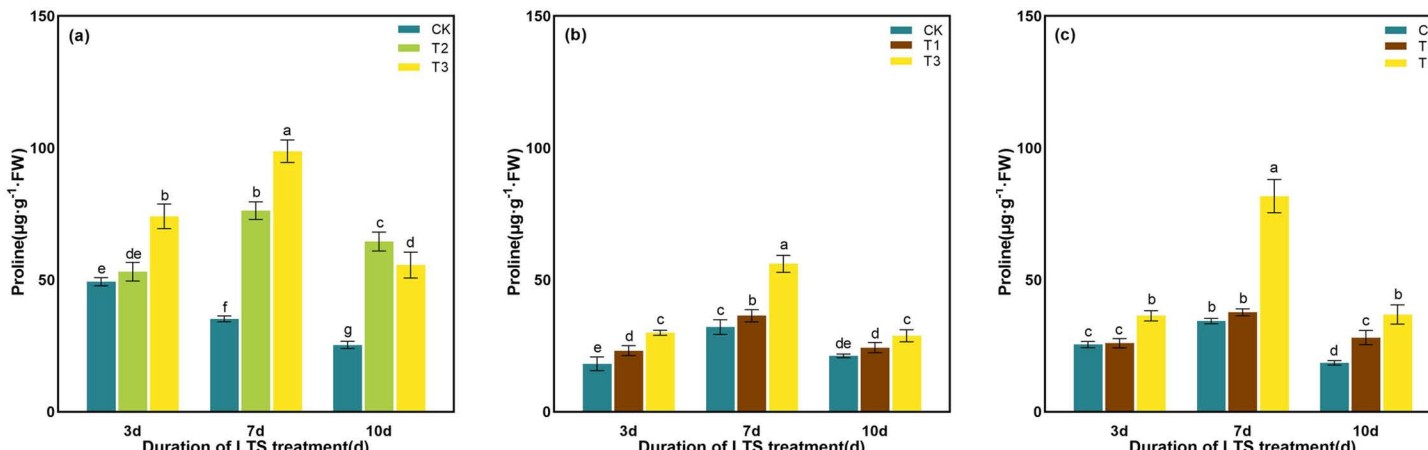

**Fig 13. Effects of LTS on proline.** (a)-(c) the variation trend of the tillering, booting and heading stages. CK, under natural conditions (control); T1, under LTS of 17.5°C; T2, under LTS of 13.5°C; T3, under LTS of 11.5°C. 3d, 7d and 10d were duration of LTS. Error bars represented Mean±SE (n=3). Different letters in lowercase indicated significant difference of the data in all treatments at $P < 0.05$.

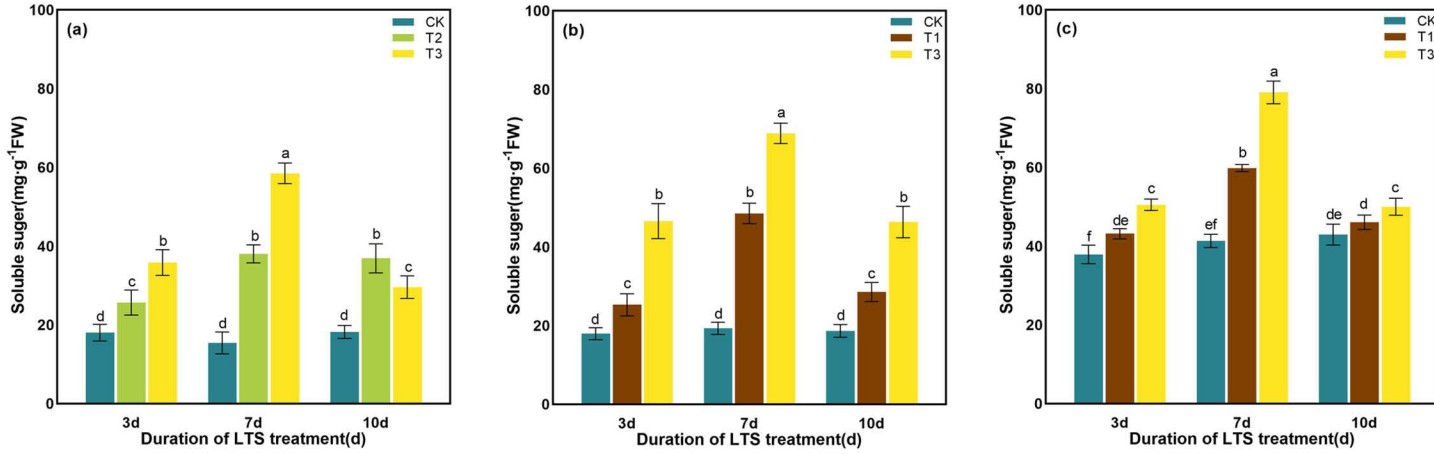

**Fig 14. Effects of LTS on the soluble sugar.** (a)-(c) the variation trend of the tillering, booting and heading stages. CK, under natural conditions (control); T1, under LTS of 17.5°C; T2, under LTS of 13.5°C; T3, under LTS of 11.5°C. 3d, 7d and 10d were duration of LTS. Error bars represented Mean±SE (n=3). Different letters in lowercase indicated significant difference of the data in all treatments at *P*<0.05.

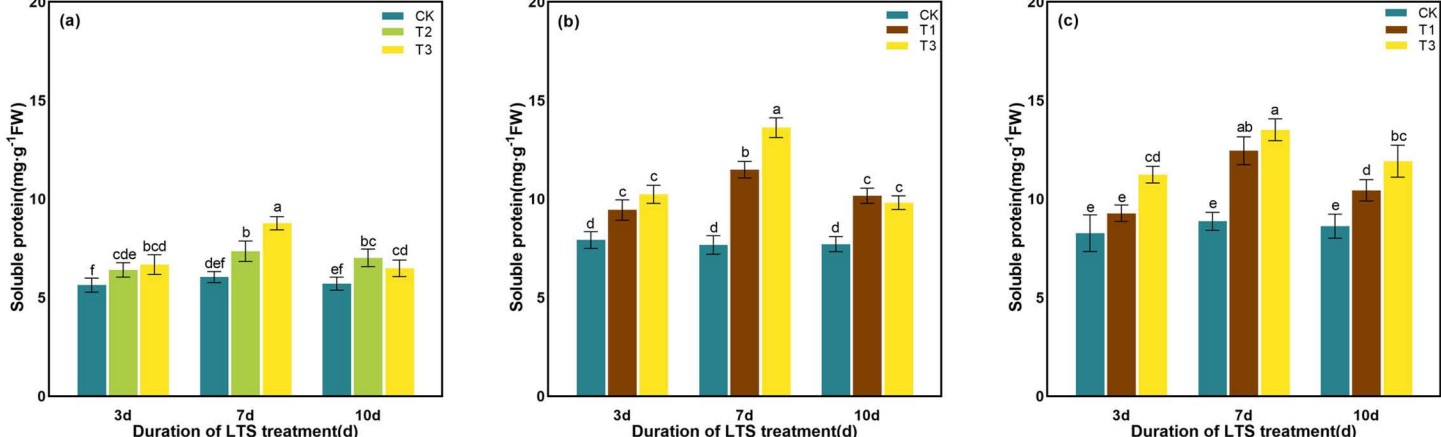

**Fig 15. Effects of LTS on the soluble protein.** (a)-(c) the variation trend of the tillering, booting and heading stages. CK, under natural conditions (control); T1, under LTS of 17.5°C; T2, under LTS of 13.5°C; T3, under LTS of 11.5°C. 3d, 7d and 10d were duration of LTS. Error bars represented Mean±SE (n=3). Different letters in lowercase indicated significant difference of the data in all treatments at *P*<0.05.

### Effect of LTS on pollen viability

With the prolongation of low-temperature treatment duration, pollen viability decreased gradually in each period (Fig 16). At the tillering stage, compared with CK, the pollen viability of T2 and T3 under LTS for 10 days decreased significantly by 9.60% and 17.27%, respectively (Fig 16a). At the booting stage, the pollen viability of D10 at T1 and T3 exhibited a significant decrease of approximately 30.04% and 31.46%, respectively, compared to the control (Fig 16b). At the heading stage, under the same LTS, there was a dramatic difference between D3 and D10, and the pollen viability of D10 at T1 and T3 exhibited a significant decrease of approximately 26.05% and 44.67%, respectively, compared to the control (Fig 16c). After 10 days of exposure to LTS, the pollen viability decreased most significantly at the heading stage, followed by the booting and the tillering stages.

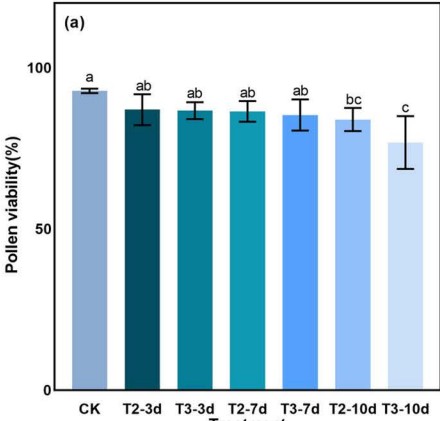 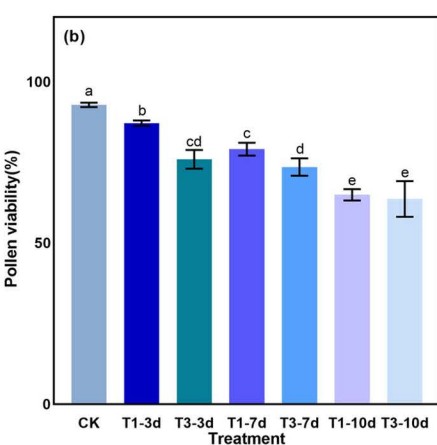 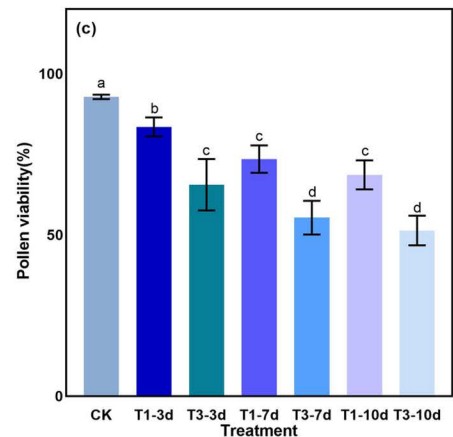

**Fig 16. Effects of LTS on pollen viability.** (a)-(c) the variation trend of the tillering, booting and heading stages. CK, under natural conditions (control); T1, under LTS of 17.5°C; T2, under LTS of 13.5°C; T3, under LTS of 11.5°C. 3d, 7d and 10d were duration of LTS. Error bars represented Mean ± SE (n = 3). Different letters in lowercase indicated significant difference of the data in all treatments at $P < 0.05$.

### Yield and its components

Grain yield and related components were significant reduced under LTS at the tillering, booting, and heading stages, Table 2. At the tillering stage, seed set percentages exhibited reductions of 2.25% and 9.80%, 7.47% and 15.21%, as well as 11.04% and 21.12% for durations of 3, 7, and 10 days at T2 and T3 in comparison to CK, with significant differences noted except for the T2 treatment at the three-day interval. Additionally, grain yield significantly decreased by 21.62% and 36.59%, 25.57% and 47.80%, as well as by 37.89% and 51.89%, corresponding to the same time frames at T2 and T3 when compared to CK. At booting stage, compared with CK, the percentage reductions observed were 29.65% and 37.70%, 30.14% and 39.34%, as well as 41.19% and 46.89% for grains per panicle, 15.48% and 60.09%, 18.72% and 77.13%, as well as 59.30% and 88.76% for grain yield, under LTS conditions T1 and T3, with durations of 3, 7, and 10 days respectively, all showing significant variations. At heading stage, under LTS of T1, grains per panicle, and grain yield were significantly decreased by 21.52%, 29.08% and 31.50%, as well as 14.69%, 36.42% and 53.01, for durations of 3, 7, and 10 days in comparison to the control, respectively. Furthermore, compared with CK, grains per panicle, and grain yield were significantly decreased by 25.13%, 34.73% and 39.51%, as well as 42.03%, 51.45% and 74.84%, for durations of 3, 7, and 10days, under LTS conditions T3, respectively.

### The correlation between $P_n$, MDA as well as yield and yield related parameters

We found a positive correlation between $P_n$ and grain yield ($p < 0.05$), effective panicles ($p < 0.05$), a negative correlation between MDA and grain yield ($p < 0.001$), effective panicles ($p < 0.001$), seed set percentage ($p < 0.001$), under LTS at tillering, booting and heading stages (Fig 17). In addition, under LTS at tillering stage, $P_n$ positively correlated with the grain ($p < 0.05$), and MDA negatively correlated with the grain ($p < 0.05$) and 1000-grain weight. Additionally, under LTS at booting and heading stages, $P_n$ positively correlated with the 1000-grain weight ($p < 0.01$) and seed set percentage ($p < 0.001$), but negatively correlated with the MDA ($p < 0.01$). The highly negative correlation of decreasing effective panicles, 1000-grain weight, and seed set percentage with MDA ($p < 0.001$) under LTS at booting and heading stages was due to significant reduction in $P_n$ ($p < 0.01$), and ultimately causing decreased grain yield.

### Discussion

As a sessile organism, rice is inherently vulnerable to environmental stressors, with LTS representing a primary constraint on its survival, growth, and yield potential. This vulnerability is particularly critical in Heilongjiang Province, the

**Table 2. Effect of LTS on grain yield and its components at different growth stages.**

| Growth stages | Treatment | Treatment days/d | Effective panicles (number/pot) | Grain/panicle | Seed set percentage (%) | 1000-grain weight (g) | Grain yield (g/pot) |
|---|---|---|---|---|---|---|---|
| Tillering stage | CK | | 17.67±2.52a | 170.67±14.50a | 89.97±1.46a | 22.81±0.27a | 37.47±6.60a |
| | T2 | 3 | 15.67±2.08a | 155.00±12.77ab | 87.95±1.61a | 22.66±0.28a | 29.37±0.85b |
| | T3 | | 13.67±4.73a | 153.67±7.02ab | 81.16±1.77b | 22.61±0.32a | 23.76±1.22bcd |
| | T2 | 7 | 14.00±2.00a | 151.33±9.02ab | 83.25±2.40b | 22.47±0.17ab | 27.89±0.98bc |
| | T3 | | 12.00±3.61a | 149.33±13.05ab | 76.29±2.31c | 22.25±0.57ab | 19.56±2.37d |
| | T2 | 10 | 13.67±0.58a | 145.67±11.50ab | 80.04±2.92b | 22.70±0.52a | 23.27±0.94 cd |
| | T3 | | 11.33±4.04a | 144.00±15.52b | 70.97±2.03d | 21.78±0.37c | 18.03±3.96d |
| Booting stage | CK | | 17.67±2.52a | 170.67±14.50a | 89.97±1.46ab | 22.81±0.27a | 37.47±6.60a |
| | T1 | 3 | 16.33±1.53a | 120.07±5.42b | 93.58±1.13a | 22.42±0.34ab | 31.67±2.74a |
| | T3 | | 14.33±2.52ab | 106.33±3.05bc | 64.29±2.99d | 21.76±0.58b | 14.95±3.85b |
| | T1 | 7 | 16.00±3.61a | 119.23±5.03b | 85.78±2.42b | 22.28±0.39ab | 30.45±5.50a |
| | T3 | | 10.33±1.53bc | 103.53±1.79 cd | 40.48±3.41e | 20.76±0.90 cd | 8.57±0.77bc |
| | T1 | 10 | 13.00±3.61ab | 100.37±7.46 cd | 70.94±4.23c | 21.51±0.48bc | 15.25±3.50b |
| | T3 | | 7.67±1.53c | 90.63±10.02d | 21.90±2.48f | 20.51±0.60d | 4.21±1.91c |
| Heading stage | CK | | 17.67±2.52a | 170.67±14.50a | 89.97±1.46a | 22.81±0.27a | 37.47±6.60a |
| | T1 | 3 | 17.33±1.53ab | 133.93±5.69b | 94.17±1.17a | 22.77±0.88a | 31.96±6.38a |
| | T3 | | 13.00±2.65c | 127.77±5.49bc | 67.98±3.63bc | 22.31±1.16ab | 21.72±5.12b |
| | T1 | 7 | 13.67±2.52bc | 121.03±7.17bcd | 73.68±2.12b | 22.60±0.55a | 23.83±1.25b |
| | T3 | | 12.33±2.52c | 111.40±8.41de | 63.19±6.88 cd | 21.52±0.93ab | 18.19±2.28b |
| | T1 | 10 | 12.00±1.00c | 116.90±6.31cde | 57.81±3.38d | 21.60±0.58ab | 17.61±2.10b |
| | T3 | | 11.33±1.53c | 103.23±9.23e | 44.88±8.37e | 20.81±1.08c | 9.43±1.71c |

Note: Within a column and within a temperature treatment, means with different lowercase letters are significantly different at P<0.05 level. CK, under the natural conditions (control); T1, LTS of 17.5°C; T2, LTS of 13.5°C; T3, LTS of 11.5°C. Mean±SE (n=3).

dominant rice-producing region in northern China. Heilongjiang serves as a vital national commodity grain base, contributing significantly to China's food security (70% of rice production as a commodity, www.zzys.moa.gov.cn). Consequently, understanding and mitigating the impact of LTS in this high-latitude, temperate zone is essential for ensuring regional productivity and stable commodity grain supplies. However, given its location in the northernmost region of China, LTS, particularly during the key development stages, poses a significant constraint on the safe production of rice [58]. Thus, Longgeng31 was selected for further comprehensive investigation into the physiology, pollen viability and yield underlying LT tolerance in rice exposed to LTS at the tillering, booting and heading stages. Longgeng31, the predominant rice variety cultivated in Heilongjiang for the last two decades, has been essential for providing economic stabilityand steady production due to its remarkable cold resilience [75]. We studied the LTS response of peroxidase, osmotic regulator, ROS, photosynthetic parameters, pollen viability and yield at tillering, booting and heading stages, providing a theoretical basis for high yield and high-quality production of LT tolerant rice breeding, and assisting rice community in taking preventative measures to control chilling damage on rice production.

## Physiological and biochemical properties of cold-region japonica rice

Low-temperature is one of the most important adversities affecting crop growth, development and yield [22]. Cold-region japonica rice plays an important role in rice production in China, and its special geographical location and climate change lead to frequent low-temperature and cold injury [14]. This study found that prolonged low-temperature exposure significantly reduced rice plant height. The stress response to low-temperature was consistent across different growth stages.

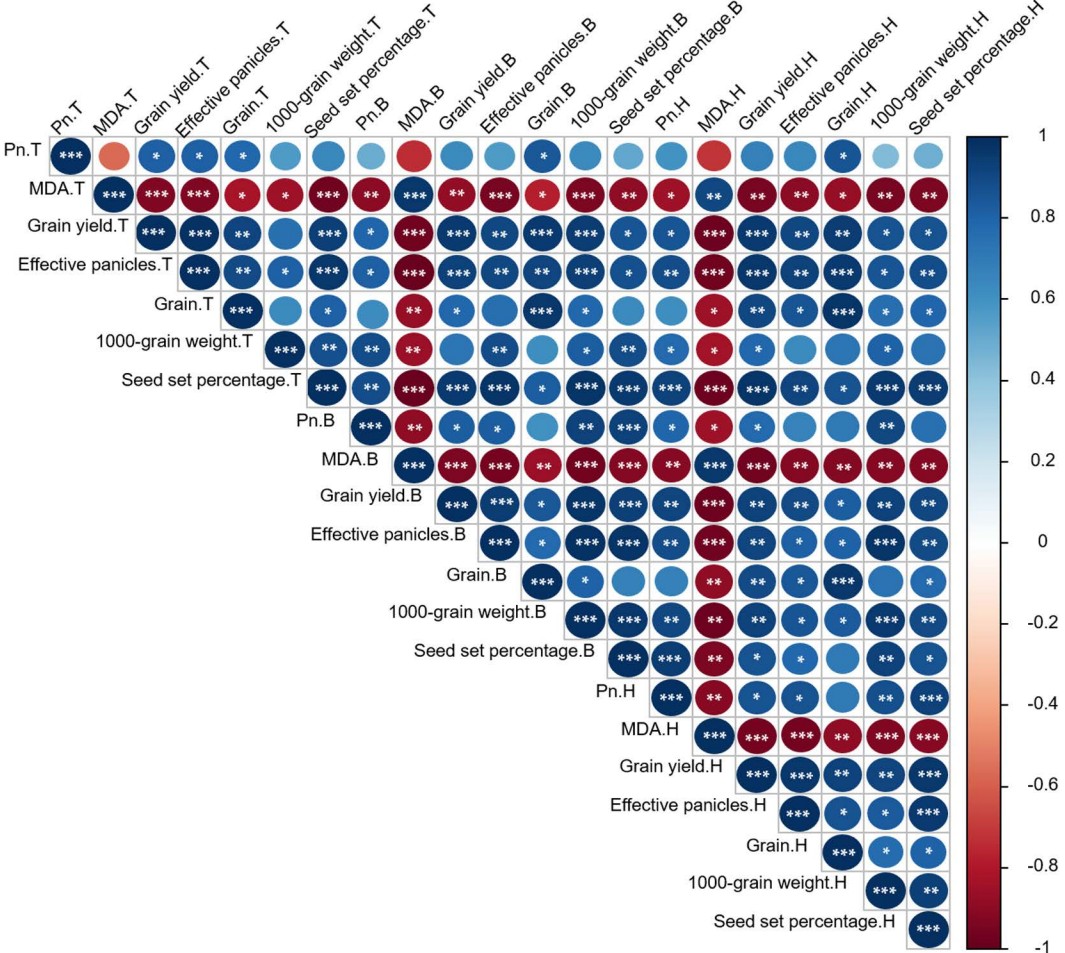

**Fig 17. The Pearson correlation matrix between Pn, MDA as well as yield and yield related parameters under LTS at tillering (T), booting (B), and heading (H) stages.** *, ** and *** represent the significant correlation at $p < 0.05$, $p < 0.01$ and $p < 0.001$, respectively.

At the same developmental stage and duration of exposure, lower temperatures resulted in a more pronounced reduction in plant height. While early-stage reductions in plant height under low-temperature conditions were not statistically significant, the extent of inhibition increased with prolonged exposure. These results indicate that extended low-temperature stress exerts a more substantial inhibitory effect on rice growth, particularly under lower temperature conditions.

Under normal temperature, plants can automatically regulate the production and removal of reactive oxygen species (ROS), so that they are in a state of dynamic equilibrium in the body. However, under low-temperature conditions, plant growth is inhibited, resulting in excessive accumulation of ROS in leaves and aggravation of membrane lipid peroxidation [69]. Plants use SOD, POD and other antioxidant enzymes to maintain the balance of active oxygen metabolism, effectively remove excess active oxygen in cells, and reduce the damage to plant cell membranes [58]. SOD, as an important active enzyme in antioxidant process, has the property of scavenging superoxide free radicals. POD is a kind of oxidoreductase produced by plants, which can eliminate the toxicity of ROS, $H_2O_2$, phenols and amines produced by plants, and reduce the damage of reactive oxygen species to plants. CAT can remove $H_2O_2$ produced by stress in plants and decompose the accumulated $H_2O_2$ into $H_2O$ and $O_2$, thereby reducing its oxidative damage to plant tissues [76]. When plants are subjected to LTS, they will also suffer from osmotic stress and water stress. The contents of some osmotic regulatory

substances with water-retaining effect, such as soluble sugar, soluble protein and proline, will increase to maintain the osmotic balance of cells and reduce the damage to crops caused by low-temperature [77]. Soluble protein content is positively correlated with the cold resistance of rice, and the higher the soluble protein content is, the stronger the cold resistance of rice [78]. In addition, soluble proteins can gather together with soluble sugars around important organelles such as chloroplasts to protect them from stress damage [78]. In this study, the activities of SOD and POD in rice leaves were significantly enhanced by LTS, and the contents of soluble protein, soluble sugar and proline were also significantly increased, indicating that rice would open its self-protection mechanism and resist the damage of LTS through its own defense system once adverse conditions were formed. The concentrations of the antioxidant and osmotic regulation systems in rice peaked after 7 days of LTS treatment during the tillering, booting, and heading stages, indicating that the stress response of rice to LTS had reached a critical threshold (Figs 10-15).

Photosynthesis is the basis of plant growth and yield formation. Under LTS, plant photosynthesis will be inhibited [79]. Photosynthesis is a process that exhibits high sensitivity to LTS during crop development, leading to a significant reduction in the $P_n$ of rice. This study demonstrated that $P_n$ decreased with increasing intensity and duration of LTS. Furthermore, the effects of LTS on $P_n$ in rice plants varied across different growth stages, with the booting stage exhibiting the most significant reduction, followed by the tillering stage and then the heading stage, which aligns with previous research findings [23,80].

The limitation of photosynthesis is generally characterized by SL and NSL [81]. Under typical conditions, the opening of stomata results in an increase in $G_s$, thereby enhancing intercellular carbon dioxide flux and facilitating a rise in the rate of photosynthesis [79]. However, the activity of chloroplasts and Rubisco decreases with the increase of the intensity and duration of LTS, which leads to a decrease in photosynthetic ability. These processes are considered as NSL of photosynthesis [82–84]. Previous studies showed that under stress conditions at different stages, the phenomenon of photosynthesis changing from being limited by SL to NSL occurred first in the first fully expanded leaf at the top [85]. Our study showed the similar results under LTS treatments, we found that $G_s$ decreased and $C_i$ increase with the decrease in LTS treatment temperatures. This indicates that the LTS-induced decreases in $P_n$ of the first fully expanded leaves were mainly caused by NSL at the tillering, booting and heading stages. This reduction was probably attributable to impaired chloroplast structures in plants under LTS, resulting in diminished capacity for $CO_2$ fixation and assimilation in the cells [22,85].

The progressive deterioration of chloroplast ultrastructure observed via TEM in rice leaves under LTS, correlating with stress intensity and duration, offered a clear structural explanation for the photosynthetic decline. This damage directly compromised critical chloroplast functions and inhibits their proper development/biogenesis, establishing a direct cytological link between low-temperature-induced structural injury within the chloroplast and the suppression of photosynthetic capacity [73]. The findings of this study showed that, compared with the CK, an increase in intensity and extended duration of LTS led to distorted chloroplast morphology, additional damage to the grana stacking structure within thylakoids, an increase in both the size and number of peroxisomes, and a loosening of stroma lamellae. These findings indicate that LTS inhibits chloroplast biosynthesis, consequently reducing the rate of photosynthesis (Fig 18).

## Effects of different growth stages LTS on yields and its components in rice

Yield attribute determines the final yield of rice, the detrimental impact of prolonged low-temperatures on rice yield has been extensively documented in the literature [23,33,86,87]. Previous studies [88,89] have demonstrated that prolonged low-temperatures during the reproductive phase exert an impact on rice panicle differentiation and flowering, resulting in a decrease in grain weight and seed setting rate, consequently leading to a reduction in rice yield. In this study, LTS reduced rice yield and its components at different growth stages, with the magnitude of decrease exhibiting significant variability. The grain yield, effective panicles, grain number per panicle, 1000-grain weight and seed setting percentage of rice decreased the most at booting stage, followed by heading stage and tillering stage, as the intensity and duration of LTS increased (Table 2). LTS during the reproductive phase significantly reduces rice yield, primarily due to a substantial increase in panicle

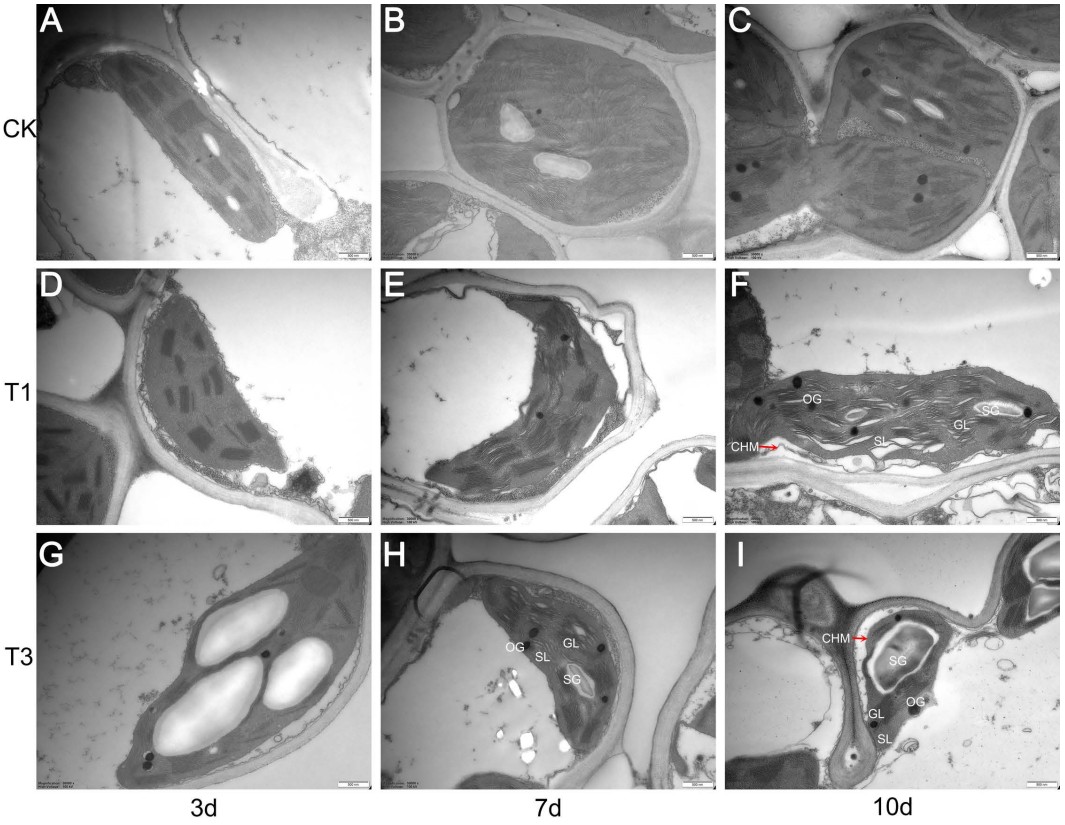

**Fig 18. The effect of LTS on chloroplast ultrastructure of rice at booting stage.** Chloroplast structure and thylakoid organization in control and LTS were analyzed by transmission electron microscopy (TEM). CK, under the natural conditions (control) for duration of 3, 7 and 10 days (A, B and **C**); T1, LTS of mean temperature 17.5°C for duration of 3, 7 and 10 days (D, E and **F**); T3, LTS of mean temperature 11.5°C for duration of 3, 7 and 10 days (G, H and **I**). G: granum; SL: stroma lamellae; GL: grana lamellae; SG: starch grain; OG: osmophilic gramules; CHM: chloroplast membrane. Bar = 500 nm.

sterility [10,46]. The results of this study revealed a significant decrease in pollen viability at the booting stage and heading stage under LTS, when compared to the CK. However, no significant changes were observed in pollen viability at the tillering stage (Figure 16). These offered further substantiation that LTS at booting and heading stage had a greater effect on seed setting rate, with a significantly higher decrease rate compared to that observed at the tillering stage. Moreover, a similar trend was observed in terms of yield. The highly positive correlation of decreasing grain yield, grain, 1000-grain weight, and seed set percentage with pollen viability ($p < 0.001$) under LTS (Fig 19), indicating that LTS inhibited pollination, causing incomplete or abnormal pollen formation, alongside decreased pollen viability and lower seed set percentages.

## Conclusions

This study showed that LTS significantly affects rice physiology, reproductive development, and yield, with the extent of damage depending on both the growth stage and the duration of exposure. The results showed that plant height and pollen activity reduced substantial under LTS. MDA, $H_2O_2$, $O_2^-$ and EL were spanking enhanced. SOD, POD and CAT activities were preeminently augmented. The contents of soluble sugar, soluble protein and proline also elevated dramatically. The $P_n$ decreased the most at the booting stage, followed by the tillering, and the heading stages. LTS decreased $P_n$ of the fully expanded flag leaf at the top during the tillering stage, booting and heading stages as a result of NSL. In response to the challenges brought by LTS, cold resistance can be improved through gene editing or transgenic technology, or by

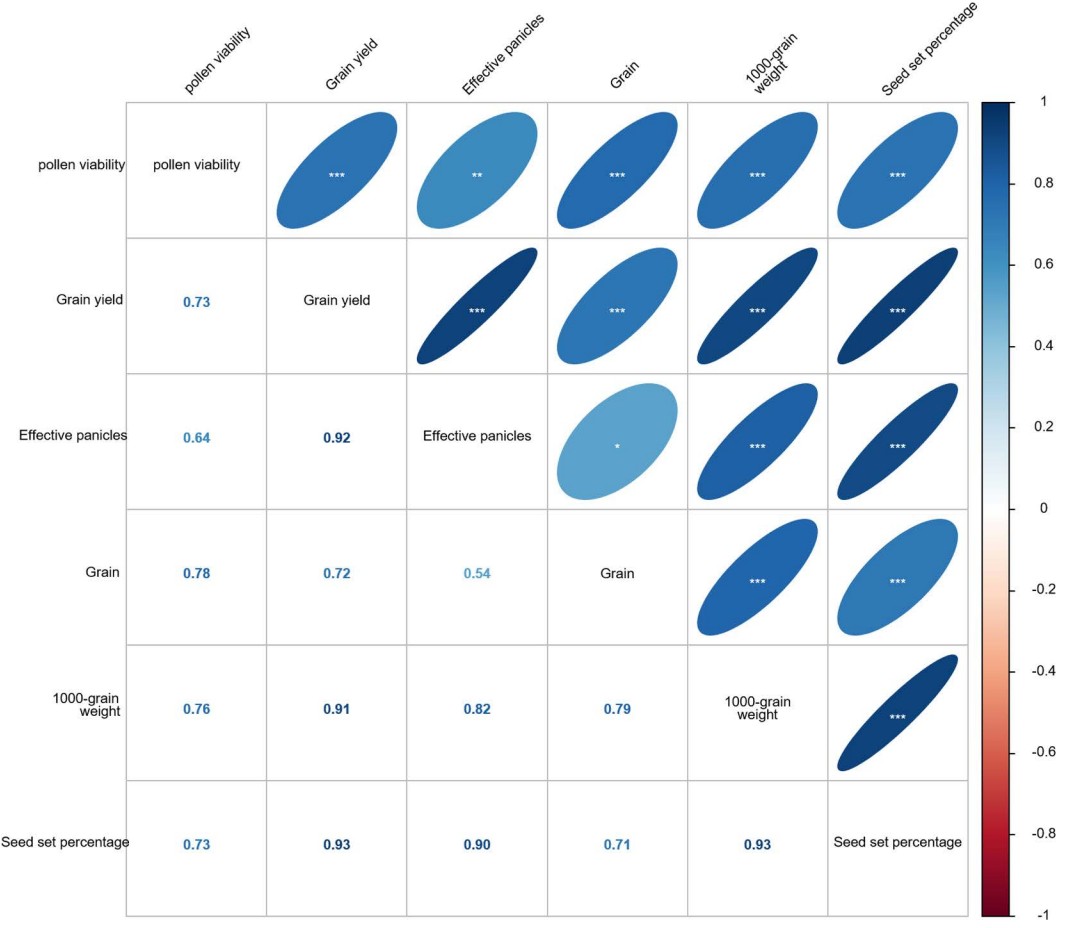

**Fig 19. The Pearson correlation matrix between pollen viability as well as yield and yield related parameters under LTS.** *, ** and *** represent the significant correlation at p<0.05, p<0.01 and p<0.001, respectively.

activating specific signal transduction pathways. These measures can effectively improve the cold resistance of rice and other crops under LTS, reduce the loss caused by low-temperature, and ensure food security.

## Supporting information

**S1 File. S1 Fig. The hourly temperatures in the phytotron.** S2 Fig. **Effects of LTS on Pn.** (a)-(c) the variation trend of the tillering, booting and heading stages. CK, under natural conditions (control); T1, under LTS of 17.5°C; T2, under LTS of 13.5°C; T3, under LTS of 11.5°C. 3d, 7d and 10d were duration of LTS. Error bars represented Mean±SE (n=3). Different letters in lowercase indicated significant difference of the data in all treatments at P<0.05. S3 Fig. **Effects of LTS on $C_i$.** (a)-(c) the variation trend of the tillering, booting and heading stages. CK, under natural conditions (control); T1, under LTS of 17.5°C; T2, under LTS of 13.5°C; T3, under LTS of 11.5°C. 3d, 7d and 10d were duration of LTS. Error bars represented Mean±SE (n=3). Different letters in lowercase indicated significant difference of the data in all treatments at P<0.05. S4 Fig. **Effects of LTS on $G_s$.** (a)-(c) the variation trend of the tillering, booting and heading stages. CK, under natural conditions (control); T1, under LTS of 17.5°C; T2, under LTS of 13.5°C; T3, under LTS of 11.5°C. 3d, 7d and 10d were duration of LTS. Error bars represented Mean±SE (n=3). Different letters in lowercase indicated significant

difference of the data in all treatments at $P<0.05$. S5 Fig. **Effects of LTS on $T_r$.** (a)-(c) the variation trend of the tillering, booting and heading stages. CK, under natural conditions (control); T1, under LTS of 17.5°C; T2, under LTS of 13.5°C; T3, under LTS of 11.5°C. 3d, 7d and 10d were duration of LTS. Error bars represented Mean±SE (n=3). Different letters in lowercase indicated significant difference of the data in all treatments at $P<0.05$. S6 Fig. **Effects of LTS on MDA.** (a)-(c) the variation trend of the tillering, booting and heading stages. CK, under natural conditions (control); T1, under LTS of 17.5°C; T2, under LTS of 13.5°C; T3, under LTS of 11.5°C. 3d, 7d and 10d were duration of LTS. Error bars represented Mean±SE (n=3). Different letters in lowercase indicated significant difference of the data in all treatments at $P<0.05$. S7 Fig. **Effects of LTS on EL.** (a)-(c) the variation trend of the tillering, booting and heading stages. CK, under natural conditions (control); T1, under LTS of 17.5°C; T2, under LTS of 13.5°C; T3, under LTS of 11.5°C. 3d, 7d and 10d were duration of LTS. Error bars represented Mean±SE (n=3). Different letters in lowercase indicated significant difference of the data in all treatments at $P<0.05$. S8 Fig. **Effects of LTS on the production rate of $O_2^-$.** (a)-(c) the variation trend of the tillering, booting and heading stages. CK, under natural conditions (control); T1, under LTS of 17.5°C; T2, under LTS of 13.5°C; T3, under LTS of 11.5°C. 3d, 7d and 10d were duration of LTS. Error bars represented Mean±SE (n=3). Different letters in lowercase indicated significant difference of the data in all treatments at $P<0.05$. S9 Fig. **Effects of LTS on $H_2O_2$.** (a)-(c) the variation trend of the tillering, booting and heading stages. CK, under natural conditions (control); T1, under LTS of 17.5°C; T2, under LTS of 13.5°C; T3, under LTS of 11.5°C. 3d, 7d and 10d were duration of LTS. Error bars represented Mean±SE (n=3). Different letters in lowercase indicated significant difference of the data in all treatments at $P<0.05$. S10 Fig. **Effects of LTS on SOD.** (a)-(c) the variation trend of the tillering, booting and heading stages. CK, under natural conditions (control); T1, under LTS of 17.5°C; T2, under LTS of 13.5°C; T3, under LTS of 11.5°C. 3d, 7d and 10d were duration of LTS. Error bars represented Mean±SE (n=3). Different letters in lowercase indicated significant difference of the data in all treatments at $P<0.05$. S11 Fig. **Effects of LTS on POD.** (a)-(c) the variation trend of the tillering, booting and heading stages. CK, under natural conditions (control); T1, under LTS of 17.5°C; T2, under LTS of 13.5°C; T3, under LTS of 11.5°C. 3d, 7d and 10d were duration of LTS. Error bars represented Mean±SE (n=3). Different letters in lowercase indicated significant difference of the data in all treatments at $P<0.05$. S12 Fig. **Effects of LTS on CAT.** (a)-(c) the variation trend of the tillering, booting and heading stages. CK, under natural conditions (control); T1, under LTS of 17.5°C; T2, under LTS of 13.5°C; T3, under LTS of 11.5°C. 3d, 7d and 10d were duration of LTS. Error bars represented Mean±SE (n=3). Different letters in lowercase indicated significant difference of the data in all treatments at $P<0.05$. S13 Fig. **Effects of LTS on proline.** (a)-(c) the variation trend of the tillering, booting and heading stages. CK, under natural conditions (control); T1, under LTS of 17.5°C; T2, under LTS of 13.5°C; T3, under LTS of 11.5°C. 3d, 7d and 10d were duration of LTS. Error bars represented Mean±SE (n=3). Different letters in lowercase indicated significant difference of the data in all treatments at $P<0.05$. S14 Fig. **Effects of LTS on the soluble sugar.** (a)-(c) the variation trend of the tillering, booting and heading stages. CK, under natural conditions (control); T1, under LTS of 17.5°C; T2, under LTS of 13.5°C; T3, under LTS of 11.5°C. 3d, 7d and 10d were duration of LTS. Error bars represented Mean±SE (n=3). Different letters in lowercase indicated significant difference of the data in all treatments at $P<0.05$. S15 Fig. **Effects of LTS on the soluble protein.** (a)-(c) the variation trend of the tillering, booting and heading stages. CK, under natural conditions (control); T1, under LTS of 17.5°C; T2, under LTS of 13.5°C; T3, under LTS of 11.5°C. 3d, 7d and 10d were duration of LTS. Error bars represented Mean±SE (n=3). Different letters in lowercase indicated significant difference of the data in all treatments at $P<0.05$. S16 Fig. **Effects of LTS on pollen viability.** (a)-(c) the variation trend of the tillering, booting and heading stages. CK, under natural conditions (control); T1, under LTS of 17.5°C; T2, under LTS of 13.5°C; T3, under LTS of 11.5°C. 3d, 7d and 10d were duration of LTS. Error bars represented Mean±SE (n=3). Different letters in lowercase indicated significant difference of the data in all treatments at $P<0.05$. S17 Fig. **The Pearson correlation matrix between Pn, MDA as Well as Yield and Yield Related Parameters under LTS at tillering (T), booting (B), and heading (H) stages.** *, ** and *** represent the significant correlation at $p<0.05$, $p<0.01$ and $p<0.001$, respectively. S18 Fig. **The effect of LTS on chloroplast ultrastructure of rice at booting stage.** Chloroplast structure and thylakoid

organization in control and LTS were analyzed by transmission electron microscopy (TEM). CK, under the natural conditions (control) for duration of 3, 7 and 10 days (A, B and C); T1, LTS of mean temperature 17.5°C for duration of 3, 7 and 10 days (D, E and F); T3, LTS of mean temperature 11.5°C for duration of 3, 7 and 10 days (G, H and I). G: granum; SL: stroma lamellae; GL: grana lamellae; SG: starch grain; OG: osmophilic gramules; CHM: chloroplast membrane. Bar = 500 nm. S19 Fig. **The Pearson correlation matrix between Pollen Viability as Well as Yield and Yield Related Parameters under LTS.** *, ** and *** represent the significant correlation at p < 0.05, p < 0.01 and p < 0.001, respectively. (PDF)

## Author contributions

**Conceptualization:** Xiaodong Du, Jianing Chang.

**Data curation:** Xiaodong Du, Zheng Chu.

**Formal analysis:** Zheng Chu.

**Funding acquisition:** Lixia Jiang.

**Investigation:** Zheng Chu.

**Methodology:** Jianing Chang, Zheng Chu.

**Project administration:** Jianing Chang.

**Resources:** Jiajia Lv.

**Software:** Jingjin Gong, Jiajia Lv.

**Supervision:** Jingjin Gong, Jiajia Lv.

**Writing – original draft:** Lifeng Guo, Lixia Jiang.

**Writing – review & editing:** Lifeng Guo, Lixia Jiang.

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
