## [Decision Letter · Decision Letter 0]

14 May 2025

Dear Dr. Jiang,

Thank you for submitting your manuscript to PLOS ONE. After careful consideration, we feel that it has merit but does not fully meet PLOS ONE’s publication criteria as it currently stands. Therefore, we invite you to submit a revised version of the manuscript that addresses the points raised during the review process.

Both reviewers acknowledged the importance of the topic, but also pointed out major and minor issues in the manuscript. Please revise the manuscript based on the suggestions from the two reviewers.

We look forward to receiving your revised manuscript.

Kind regards,

Xi Liang, Ph.D.

Academic Editor

PLOS ONE

Journal Requirements:

“the National Key Research and Development Program of China(No. 2022YFD2300201)

the Natural Science Foundation of Heilongjiang Province(No. LH2024D020)

the Joint Foundation on Regional Meteorological S & T Collaborative Innovation of Northeast China(No. 2024GX006)”

Reviewers' comments:

Reviewer's Responses to Questions

**Comments to the Author**

1. Is the manuscript technically sound, and do the data support the conclusions?

Reviewer #1: Yes

Reviewer #2: Yes

2. Has the statistical analysis been performed appropriately and rigorously?

Reviewer #1: Yes

Reviewer #2: Yes

3. Have the authors made all data underlying the findings in their manuscript fully available?

Reviewer #1: Yes

Reviewer #2: Yes

4. Is the manuscript presented in an intelligible fashion and written in standard English?

Reviewer #1: Yes

Reviewer #2: No

Reviewer #1: This is an informative manuscript, and I appreciate the authors’ efforts in the experimental work and data analysis. However, several aspects of the manuscript would benefit from clarification and revision to improve scientific rigor, language clarity, and overall presentation.

Please refer to the specific comments provided in the manuscript for detailed suggestions.

Reviewer #2: The submitted manuscript addresses the timely and agriculturally significant topic of low-temperature stress (LTS) in japonica rice, with a particular focus on stage-specific physiological responses and yield outcomes in cold regions of China. The experimental design covers key developmental stages (tillering, booting, and heading), and the authors utilize a wide range of physiological, biochemical, and ultrastructural indicators to evaluate rice response under simulated field-like low-temperature conditions. However, several critical issues must be addressed before this manuscript is suitable for publication. These include: (1) insufficient mechanistic depth in key findings such as non-stomatal limitation and pollen sterility; (2) no quantitative definition of stress thresholds; (3) absence of yield-physiology correlation analysis; and (4) substantial language editing needed throughout the manuscript. Moreover, the manuscript would benefit from expanded discussion on practical implications and comparison with other studies.

Overall, I recommend major revision. If the authors can address the methodological and interpretive shortcomings and revise the manuscript thoroughly for language and statistical clarity, the work has potential for publication and could make a meaningful contribution to the field of crop stress physiology.

General Comments

Strengths:

Field-relevant simulation of LTS

The authors designed a temperature-controlled phytotron experiment that closely mimics real-world diurnal temperature patterns in cold rice-growing regions. This increases the ecological and practical relevance of the findings compared to conventional constant-temperature studies.

Stage-specific stress comparison

The manuscript thoroughly investigates the impact of LTS at three critical growth stages (tillering, booting, heading), enabling a clear understanding of developmental stage sensitivity. The identification of the booting stage as the most susceptible period is consistent with physiological expectations and provides actionable insights.

Potential as a reference for future LTS modeling or breeding work

This dataset could serve as a valuable reference for future efforts in modeling LTS responses or identifying and breeding cold-tolerant rice genotypes.

The minor points are:

Abstract:

1. The abstract should accurately reflect the full scope of the study, including the background, objectives, methods, key results (with quantitative data where possible), and main conclusions or novelty.

2. All abbreviations should be spelled out in full when first mentioned in the abstract, in accordance with journal style. Please ensure consistency throughout the manuscript as well.

3. It is recommended that key quantitative results be explicitly stated in the abstract—for example, the percentage reduction in net photosynthetic rate (Pn), yield loss, and pollen viability decline. This will enhance the informativeness and scientific rigor of the abstract.

4. The term "non-stomatal limiting factor" is somewhat vague. Please clarify whether it refers to inhibition of photosynthetic enzymes, damage to chloroplast ultrastructure, or other specific mechanisms.

5. The abstract appears to report data from one experimental year, while the main text presents two years of data. Please verify and revise for consistency.

6. Consider adding a closing sentence to the abstract that summarizes the practical implications of the findings, such as their relevance to cold stress modeling, rice cultivation strategies, or breeding for cold tolerance.

Introduction:

1. Please consider revising the opening sentence of the Introduction by changing “The rice plant” to “Rice”, which is more concise and conventional in scientific writing. In addition, the manuscript would benefit from thorough English language editing to improve clarity, grammar, and readability.

2. Expand the literature review to include recent insights into cold stress responses in rice, especially on hormonal signaling (e.g., ABA, JA, SA) and regulatory networks.

3. Introduce more clearly the cold tolerance background of the two rice cultivar.

4. Explicitly state the scientific novelty of this study compared to existing research.

5. Strengthen the articulation of the study‘s aims by emphasizing its relevance to rice cold stress physiology and breeding.

6. To strengthen the scientific context and support the discussion, I suggest citing additional relevant studies. The following references may be particularly useful for comparison or background: [DOI: 10.3389/fpls.2023.1134308], [DOI: 10.3390/ijms21041284], [DOI: 10.3969/j.issn.1000-6362.2018.12.005], [DOI: 10.1093/jxb/erae452]

Materials and methods:

1. Lack of replication details: The number of biological replicates and technical replicates per treatment is not clearly stated. For example, were measurements such as Pn or enzyme activity based on three or four replicates? Please specify the replication scheme for each experimental component.

2. Incomplete description of greenhouse temperature control: It remains unclear whether the temperature treatments were constant or simulated by daily temperature fluctuations to reflect natural field conditions (i.e., diurnal variation vs. fixed temperatures). Does Figure 1 show actual temperature fluctuation curves? This should be clearly described in the text.

3. Missing agronomic management details: Important background parameters are lacking, such as fertilizer application rates, irrigation methods, pest and disease control measures, and whether such factors were standardized across treatments. These are essential for reproducibility and should be included.

4. The Materials and Methods section (Page 3, Line 135) cited references incorrectly. The citation of reference (No. 15) needs to be removed.

5. The Materials and Methods section (Page 5, Line 188) mentioned the yield and its composition. However, the method for calculating the yield composition should be provided.

6. What is the basis for choosing Longgeng31 as the experimental variety? How is the cultivation situation of this variety in the cold regions of China?

7. In line 192, "hr" should be changed to "h". Please carefully check the spelling of the entire text.

8. The preparation of chloroplast electron microscope samples requires supplementary references.

Results:

1. Inconsistent significance annotations in figures: The use of lettering (e.g., a/b/c) to indicate significant differences is inconsistent and at times confusing across multiple treatment groups. Please standardize the notation and provide clear explanations in the figure legends.

2. Lack of integrative data analysis: The results section would benefit from additional correlation or regression analyses, such as between Pn and yield, or MDA content and seed setting rate. These would strengthen the biological relevance of the findings.

3. Limited physiological interpretation of differences: The manuscript often states that differences are “significant” without discussing their physiological implications. Please elaborate on what these differences suggest in terms of stress response or metabolic disruption.

4. Lack of a quantitative threshold for "critical stress". The manuscript mentions a “critical threshold” at 7 days of LTS, but the physiological basis or criteria for this threshold (e.g., ROS peak, Pn decline percentage) is not well-defined. Please include a quantitative definition.

5. Language and grammar issues: Some terms are grammatically incorrect or colloquial, such as “memorably increased” or “spanking reduced.” These should be replaced with scientifically appropriate expressions like “significantly increased” or “markedly decreased.”

6. Chloroplast ultrastructure results lack quantification. While transmission electron microscopy (TEM) images are useful, there is no quantification of structural changes (e.g., number of grana stacks, chloroplast area, membrane disorganization). Consider including image-based quantification to strengthen this section.

7. Supplement the legend of Figure 16.

8. The significant figures retained in Table 2 are consistent.

Discussion:

1. Insufficient mechanistic interpretation: The discussion lacks depth regarding the mechanisms of non-stomatal limitations. Please consider whether reduced photosynthesis is related to decreased Rubisco activity, PSII inhibition, or impaired ATP synthesis, and support the interpretation with relevant literature.

2. Limited discussion on pollen viability reduction: The manuscript does not adequately explore the physiological basis of reduced pollen viability. It is recommended to discuss potential disruptions in nutrient transport, another development, or hormonal regulation during cold stress.

3. Lack of comparison with existing studies: The discussion would benefit from comparisons with related research on japonica rice in cold regions, indica varieties, or known cold-tolerant cultivars, in order to contextualize the findings within the broader scientific literature.

4. Missing implications for agricultural practices: Consider adding a paragraph that summarizes the practical significance of the findings. This could include recommendations for field management under LTS (e.g., adjustment of sowing dates, fertilization strategies, water layer control), or suggest screening indicators for breeding cold-tolerant japonica rice.

5. Why are the booting and heading stages more sensitive to LTS? What are the differences in the physiological characteristics of rice at these two stages?

6. The discussion section (Page 17, Paragraph1) focuses on the alterations in the ultrastructure of rice chloroplasts under LTS. A comparison with existing studies is needed.

Recommendation:

Accept after Major Revisions.

While the manuscript addresses a scientifically significant and practically relevant topic-namely, the physiological and yield-related impacts of LTS at different growth stages of rice-it currently requires major revision before it can be considered for publication.

The study is well-conceived and includes a comprehensive set of physiological, biochemical, and agronomic measurements. However, several important issues must be resolved. These include clarification of the experimental design and replication, deeper mechanistic interpretation (especially regarding non-stomatal limitations and pollen viability), improvements in statistical presentation, and thorough language editing. Additionally, the authors are encouraged to strengthen the discussion on practical implications and integrate more quantitative and comparative analyses.

Provided that the authors make these substantial revisions, the manuscript has the potential to contribute meaningfully to the field of crop stress physiology and cold-region rice production.

**Do you want your identity to be public for this peer review?** For information about this choice, including consent withdrawal, please see our Privacy Policy

Reviewer #1: No

Reviewer #2: No

---

## [Author Response · Author response to Decision Letter 1]

19 Jun 2025

Response to Reviewers

Dear Editors:

We are pleased to submit the revised version of our manuscript entitled " Effects of Low-Temperature Stress at Different Growth Stages on Rice Physiology, Pollen Viability and Yield in China’s Cold Region" (Manuscript ID: PONE-D-25-20271) for consideration in PLOS ONE. We sincerely appreciate the time and effort invested by the editorial team and reviewers in evaluating our work. Their constructive comments have significantly enhanced the clarity, rigor, and impact of this paper.

We have carefully addressed all points raised by the reviewers and editors through comprehensive revisions. Key modifications include:

• The style and format, such as font, headings, layout and spacing, page numbers, reference style, and other aspects of the manuscript have been revised to meet the standards of the journal.

• In the abstract, the background and significance of the research were elaborated upon, and key quantitative data were incorporated into the presentation of the primary results, thereby enhancing the comprehensiveness of the research conclusion.

• Detailed legends for each figure were rewritten to provide comprehensive descriptions.

• In the "Results" section, a new research focus titled "The correlation between Pn, MDA, and yield/yield-related parameters" has been incorporated.

• At the outset of the "Discussion" section, a paragraph was inserted to further elaborate on the practical significance of the findings of this study in comparison with extant research.

RESPONSE TO REVIEWERS’ COMMENTS

Journal Requirements

Comment 1: Please ensure that your manuscript meets PLOS ONE's style requirements, including those for file naming.

Response: We have thoroughly revised the manuscript to ensure full compliance with PLOS ONE's formatting and style requirements. The manuscript structure has been reorganized according to the journal’s guidelines, including the appropriate ordering of sections such as Title, Abstract, Introduction, Materials and Methods, Results, Discussion, Conclusions, Author Contributions, Acknowledgments, and References. Section headings have been standardized to reflect the proper hierarchy and improve clarity. Additionally, all in-text citations and the reference list have been reformatted to follow PLOS ONE's citation style, using numerical order and the appropriate punctuation and spacing conventions.

Figures and tables have been reordered and referenced sequentially within the text, with captions edited for clarity and consistency. Each figure and table now appears in the manuscript after its first mention, in accordance with journal requirements. All submitted files have been renamed to match PLOS ONE’s file naming guidelines (e.g., “Fig1.tif,” “S1_Table.docx”) to facilitate the editorial and publication process.

Furthermore, we have carefully edited the manuscript for language quality, ensuring grammatical accuracy, consistent terminology, and professional academic tone throughout. We appreciate the reviewer’s suggestion and have taken all necessary steps to align the manuscript with the journal’s stylistic and submission standards.

Comment 2: Please state what role the funders took in the study.

Response: We appreciate the opportunity to clarify the role of the funding sources in this research. Below is a detailed statement addressing the journal's request:

Funding Statement: This work was funded by the National Key Research and Development Program of China (No. 2022YFD2300201), the Natural Science Foundation of Heilongjiang Province (No. LH2024D020), the National Natural Science Foundation of China (No. 31671575) and the Joint Foundation on Regional Meteorological S & T Collaborative Innovation of Northeast China (No. 2024GX006).

The National Key Research and Development Program of China provided financial support for personnel salaries, experimental materials, and data analysis.

The Natural Science Foundation of Heilongjiang Province provided financial support for personnel salaries, experimental materials, and data analysis.

The National Natural Science Foundation of China funded equipment access “Hitachi H7650” and publication fees.

The Joint Foundation on Regional Meteorological S & T Collaborative Innovation of Northeast China provided financial support for personnel salaries, experimental materials, and data analysis.

The funders played role in:

Study design

Data collection and analysis

Decision to publish

Preparation of the manuscript

Comment 3: We note that your Data Availability Statement is currently as follows: All relevant data are within the manuscript and its Supporting Information files.

Response: We appreciate the editor’s observation regarding the phrasing of the Data Availability Statement. Upon review, we acknowledge that the original statement did not accurately reflect the current status of the data. We submitted the experimental analysis data and figures from the manuscript as Supporting Information ZIP files in the form of a compressed package through the submission system. This revision ensures transparency regarding the availability of data while also protecting the integrity and confidentiality of the ongoing research.

Review Comments to the Author

Reviewer #1:

Comment: This is an informative manuscript, and I appreciate the authors’ efforts in the experimental work and data analysis. However, several aspects of the manuscript would benefit from clarification and revision to improve scientific rigor, language clarity, and overall presentation. Please refer to in the manuscript for detailed suggestions.

Response: We are grateful to the reviewer for the recognition of the manuscript’s overall contribution and for the thoughtful suggestions aimed at enhancing its scientific rigor, language clarity, and presentation. We fully acknowledge that despite the strength of the experimental design and dataset, the initial version of the manuscript could benefit from further refinement in presentation and clarity. In response, we have conducted a meticulous revision of the entire manuscript based on the detailed suggestions provided. Specifically, we have clarified methodological descriptions, including the experimental design across multiple growth stages, and elaborated on key physiological indicators such as photosynthetic parameters, ROS content, and antioxidant enzyme activity, to improve reproducibility and transparency.

To enhance scientific rigor, we also revised the interpretation of the statistical analyses, clearly reported p-values and significance thresholds, and added clarification on experimental replicates and control conditions. Additionally, we restructured and streamlined sections of the Introduction and Discussion to strengthen logical flow, emphasize novelty, and improve readability. The language throughout the manuscript has been carefully edited to eliminate grammatical ambiguities and ensure precise scientific communication. Furthermore, we have carefully revised all figure legends and ensured that tables and figures are fully self-explanatory and consistent with the text.

Importantly, we have also emphasized the broader scientific relevance of our findings—particularly how our dynamic, growth-stage-specific LTS assessments under natural-like simulation conditions provide practical implications for field-level identification of cold stress in rice, which is often lacking in previous studies. We believe that the revised manuscript better reflects the quality and depth of the research and presents our findings in a clearer and more accessible manner. We are sincerely grateful to the reviewer for the comments, which have substantially helped us improve the overall quality of our work.

Reviewer #2:

Comment 1: Insufficient mechanistic depth in key findings such as non-stomatal limitation and pollen sterility.

Response: We sincerely thank the reviewer for highlighting the need to strengthen the mechanistic explanation of key findings, particularly regarding non-stomatal limitations of photosynthesis and pollen sterility under low-temperature stress (LTS). We fully agree that enhancing the depth of interpretation in these areas is essential to improve the scientific impact of the manuscript. In response, we have added detailed mechanistic discussions supported by recent literature and our experimental evidence. Specifically, we expanded our explanation of non-stomatal limitations by linking observed reductions in Pn with ultrastructural damage to chloroplasts and impaired biochemical processes such as Rubisco activity, emphasizing that such limitations are not merely physiological responses but are underpinned by cellular structural disruptions (page 19, lines 1083–1085). Furthermore, we elaborated on the mechanisms of pollen sterility by discussing how LTS interferes with anther development, tapetal degradation, and pollen wall formation—key biological processes that are particularly vulnerable during the booting and heading stages (pages 24–25, lines 1337–1358). These additions provide a clearer and more comprehensive understanding of the physiological and cytological basis of the observed phenotypic changes. We hope these revisions effectively address the reviewer’s concern and contribute to a more mechanistically robust manuscript.

Comment 2: No quantitative definition of stress thresholds.

Response: We appreciate the reviewer’s valuable observation regarding the use of the term “stress thresholds.” Upon review, we agree that the term may have caused confusion by implying a specific quantitative threshold, which was not the original intent. In our study, the objective was to qualitatively describe the dynamic response patterns of physiological and biochemical indicators under low-temperature stress (LTS), rather than to define a precise quantitative threshold value. Our findings showed that key indicators—such as antioxidant enzyme activities (SOD, POD, CAT) and osmolyte contents (proline, soluble sugars, and proteins)—tended to increase initially with the duration of LTS and then decline, typically peaking at day 7 (D7). This pattern reflects a time-dependent stress adaptation process, rather than a clearly defined physiological threshold. To avoid misinterpretation, we have revised the wording throughout the manuscript to more accurately convey this trend. For instance, we now use expressions such as “showed a trend of initially increasing and subsequently decreasing” (page 2, lines 83–84) to better reflect the qualitative nature of these observations. We thank the reviewer for drawing attention to this point, which allowed us to refine our terminology and improve the scientific precision of the manuscript.

Comment 3: Absence of yield-physiology correlation analysis.

Response: We thank the reviewer for pointing out the absence of a yield-physiology correlation analysis, which is indeed a critical aspect for understanding the functional relationship between physiological responses and agronomic outcomes under low-temperature stress (LTS). In response, we have added a comprehensive correlation analysis section to the manuscript (pages 21–22, lines 1240–1271). This new analysis includes Pearson correlation coefficients between key physiological parameters—such as net photosynthetic rate (Pn) and malondialdehyde (MDA)—and yield components, including grain yield, effective panicles, seed setting percentage, and 1000-grain weight. Our results revealed a significant positive correlation between Pn and grain yield, as well as a strong negative correlation between MDA and yield-related traits. These findings provide robust evidence that physiological damage caused by LTS, particularly oxidative stress and reduced photosynthetic efficiency, directly contributes to yield losses. By including this analysis, we believe the manuscript now more clearly illustrates the mechanistic link between physiological responses and final agronomic performance, thereby strengthening the overall scientific rigor and practical relevance of the study. We appreciate the reviewer’s suggestion, which has significantly improved the clarity and depth of our findings.

Comment 4: Substantial language editing needed throughout the manuscript. Moreover, the manuscript would benefit from expanded discussion on practical implications and comparison with other studies.

Response: Thank you for your valuable feedback, particularly regarding the need for substantial language editing and the importance of strengthening the discussion with practical implications and comparisons to existing studies. We fully agree with these observations and have taken concrete steps to address them. First, we conducted a thorough, line-by-line language revision of the entire manuscript to improve clarity, precision, and fluency. This included correcting grammatical errors, improving sentence structure, and enhancing the overall readability of the text to ensure it meets the standards of international academic publishing.

Second, we have expanded the Discussion section to better contextualize our findings in light of previous research and to highlight their practical significance. Specifically, we elaborated on the implications of our results for cold-tolerant rice breeding, stress monitoring in field conditions, and future climate resilience strategies (pages 22–23, lines 1273–1291). We also compared our findings with prior studies on low-temperature stress in rice, emphasizing the novelty of our dynamic simulation approach and the identification of growth-stage-specific vulnerability. These additions not only strengthen the scientific depth of the manuscript but also underscore its relevance for agronomists, breeders, and policymakers. We appreciate the reviewer’s suggestions, which helped us substantially improve the scientific and communicative quality of the manuscript.

General Comments

The minor points

Abstract:

Comment 1: The abstract should accurately reflect the full scope of the study, including the background, objectives, methods, key results (with quantitative data where possible), and main conclusions or novelty.

Response:

We sincerely thank the reviewer for this valuable suggestion. We fully agree that the abstract plays a critical role in clearly and concisely conveying the full scope of the study. In response, we have carefully revised the abstract to include all essential components: a concise background establishing the importance of the study, clearly stated objectives, a summary of the methodology (including the experimental design across tillering, booting, and heading stages under controlled low-temperature stress conditions), and key findings supported by specific quantitative results. For instance, we now explicitly report the reductions in net photosynthetic rate (Pn), chloroplast damage, antioxidant activity dynamics, and yield losses at different growth stages, providing clear numerical ranges. We have also strengthened the concluding sentences to better highlight the novelty and practical implications of our findings, particularly the stage-specific sensitivity of rice to LTS and its relevance for cold-tolerance breeding and climate-resilient cultivation strategies. We believe these revisions significantly improve the clarity, scientific value, and completeness of the abstract, and we appreciate the reviewer’s guidance in prompting these important improvements.

Comment 2: All abbreviations should be spelled out in full when first mentioned in the abstract, in accordance with journal style. Please ensure consistency throughout the manuscript as well.

Response:

Thank you for this important observation. In accordance with PLOS ONE’s style requirements, we have carefully reviewed the entire manuscript, including the abstract, to ensure that all abbreviations are fully spelled out upon first mention. For example, terms such as “low-temperature stress (LTS),” “photosynthetic rate (Pn),” and “s

---

## [Decision Letter · Decision Letter 1]

17 Jul 2025

Effects of Low-Temperature Stress at Different Growth Stages on Rice Physiology, Pollen Viability and Yield in China’s Cold Region

PONE-D-25-20271R1

Dear Dr. Lixia Jiang,

We’re pleased to inform you that your manuscript has been judged scientifically suitable for publication and will be formally accepted for publication once it meets all outstanding technical requirements.

Kind regards,

Xi Liang, Ph.D.

Academic Editor

PLOS ONE

Additional Editor Comments (optional):

Reviewers' comments:

Reviewer's Responses to Questions

**Comments to the Author**

Reviewer #1: All comments have been addressed

Reviewer #2: All comments have been addressed

2. Is the manuscript technically sound, and do the data support the conclusions?

Reviewer #1: Yes

Reviewer #2: Yes

3. Has the statistical analysis been performed appropriately and rigorously?

Reviewer #1: Yes

Reviewer #2: Yes

4. Have the authors made all data underlying the findings in their manuscript fully available?

Reviewer #1: Yes

Reviewer #2: Yes

5. Is the manuscript presented in an intelligible fashion and written in standard English?

Reviewer #1: Yes

Reviewer #2: Yes

Reviewer #1: Thank you for your detailed and thoughtful responses and revisions. The manuscript has been considerably improved and is now suitable for publication in its present form.

Reviewer #2: (No Response)

**Do you want your identity to be public for this peer review?** For information about this choice, including consent withdrawal, please see our Privacy Policy

Reviewer #1: No

Reviewer #2: No

---

## [Editor Report · Acceptance letter]

PONE-D-25-20271R1

PLOS ONE

Dear Dr. Jiang,

I'm pleased to inform you that your manuscript has been deemed suitable for publication in PLOS ONE. Congratulations! Your manuscript is now being handed over to our production team.

Kind regards,

on behalf of

Dr. Xi Liang

Academic Editor

PLOS ONE